# Linguistic processing of task-irrelevant speech at a cocktail party

Paz Har-shai Yahav*, Elana Zion Golumbic*

The Gonda Center for Multidisciplinary Brain Research, Bar Ilan University, Ramat Gan, Israel

**Abstract** Paying attention to one speaker in a noisy place can be extremely difficult, because to-be-attended and task-irrelevant speech compete for processing resources. We tested whether this competition is restricted to acoustic-phonetic interference or if it extends to competition for linguistic processing as well. Neural activity was recorded using Magnetoencephalography as human participants were instructed to attend to natural speech presented to one ear, and task-irrelevant stimuli were presented to the other. Task-irrelevant stimuli consisted either of random sequences of syllables, or syllables structured to form coherent sentences, using hierarchical frequency-tagging. We find that the phrasal structure of structured task-irrelevant stimuli was represented in the neural response in left inferior frontal and posterior parietal regions, indicating that selective attention does not fully eliminate linguistic processing of task-irrelevant speech. Additionally, neural tracking of to-be-attended speech in left inferior frontal regions was enhanced when competing with structured task-irrelevant stimuli, suggesting inherent competition between them for linguistic processing.

*For correspondence:
pazhs10@gmail.com (PH-Y);
elana.zion-golumbic@biu.ac.il
(EZG)

Competing interests: The authors declare that no competing interests exist.

## Introduction

The seminal speech-shadowing experiments conducted in the 50 s and 60 s set the stage for studying one of the primary cognitive challenges encountered in daily life: how do our perceptual and linguistic systems deal effectively with competing speech inputs? (*Cherry, 1953*; *Broadbent, 1958*; *Treisman, 1960*). Over the past decades, a wealth of behavioral and neural evidence has accumulated showing that when only one speech-stream is behaviorally relevant, auditory and linguistic resources are devoted to its preferential encoding at the expense of other task-irrelevant input. Consequentially, this so-called 'attended' message can be repeated, comprehended, and remembered, whereas very little of competing task-irrelevant speech is explicitly recalled (*Glucksberg, 1970*; *Ambler et al., 1976*; *Neely and LeCompte, 1999*; *Oswald et al., 2000*; *Brungart et al., 2001*). This attentional selection is accompanied by attenuation of speech-tracking for unattended speech in auditory regions (*Mesgarani and Chang, 2012*; *Horton et al., 2013*; *Zion Golumbic et al., 2013a*; *O'Sullivan et al., 2015*; *Fiedler et al., 2019*; *Teoh and Lalor, 2019*) as well as language-related regions (*Zion Golumbic et al., 2013b*), particularly for linguistic-features of the speech (*Brodbeck et al., 2018a*; *Broderick et al., 2018*; *Ding et al., 2018*; *Brodbeck et al., 2020a*). However, as demonstrated even in the earliest studies, the content of task-irrelevant speech is probably not fully suppressed and can affect listener behavior in a variety of ways (*Moray, 1959*; *Bryden, 1964*; *Yates, 1965*). Indeed, despite decades of research, the extent to which concurrent speech are processed and the nature of the competition for resources between 'attended' and 'task-irrelevant' input in multi-speaker contexts is still highly debated (*Kahneman, 1973*; *Driver, 2001*; *Lachter et al., 2004*; *Bronkhorst, 2015*).

Fueling this debate are often-conflicting empirical findings regarding whether or not task-irrelevant speech is processed for semantic and linguistic content. Many studies fail to find behavioral or neural evidence for processing task-irrelevant speech beyond its acoustic features (*Carlyon et al.,*

**eLife digest** We are all familiar with the difficulty of trying to pay attention to a person speaking in a noisy environment, something often known as the 'cocktail party problem'. This can be especially challenging when the background noise we are trying to filter out is another conversation that we can understand. In order to avoid being distracted in these kinds of situation, we need selective attention, the cognitive process that allows us to attend to one stimulus and to ignore other irrelevant sensory information. How the brain processes the sounds in our environment and prioritizes them is still not clear.

One of the central questions is whether we can take in information from several speakers at the same time or whether we can only understand speech from one speaker at a time. Neuroimaging techniques can shed light on this matter by measuring brain activity while participants listen to competing speech stimuli, helping researchers understand how this information is processed by the brain.

Now, Har-Shai Yahav and Zion Golumbic measured the brain activity of 30 participants as they listened to two speech streams in their native language, Hebrew. They heard each speech in a different ear and tried to focus their attention on only one of the speakers. Participants always had to attend to natural speech, while the sound they had to ignore could be either natural speech or unintelligible syllable sequences. The activity of the brain was registered using magnetoencephalography, a non-invasive technique that measures the magnetic fields generated by the electrical activity of neurons in the brain.

The results showed that unattended speech activated brain areas related to both hearing and language. Thus, unattended speech was processed not only at the acoustic level (as any other type of sound would be), but also at the linguistic level. In addition, the brain response to the attended speech in brain regions related to language was stronger when the competing sound was natural speech compared to random syllables. This suggests that the two speech inputs compete for the same processing resources, which may explain why we find it difficult to stay focused in a conversation when there are other people talking in the background.

This study contributes to our understanding on how the brain processes multiple auditory inputs at once. In addition, it highlights the fact that selective attention is a dynamic process of balancing the cognitive resources allocated to competing information rather than an all-or-none process. A potential application of these findings could be the design of smart devices to help individuals focus their attention in noisy environments.

---

*2001*; *Lachter et al., 2004*; *Ding et al., 2018*). However, others are able to demonstrate that at least some linguistic information is gleaned from task-irrelevant speech. For example, task-irrelevant speech is more distracting than non-speech or incomprehensible distractors (*Rhebergen et al., 2005*; *Iyer et al., 2010*; *Best et al., 2012*; *Gallun and Diedesch, 2013*; *Carey et al., 2014*; *Kilman et al., 2014*; *Swaminathan et al., 2015*; *Kidd et al., 2016*), and there are also indications for implicit processing of the semantic content of task-irrelevant speech, manifest through priming effects or memory intrusions (*Tun et al., 2002*; *Dupoux et al., 2003*; *Rivenez et al., 2006*; *Beaman et al., 2007*; *Carey et al., 2014*; *Aydelott et al., 2015*; *Schepman et al., 2016*). Known as the 'Irrelevant Sound Effect' (ISE), these are not always accompanied by explicit recall or recognition (*Lewis, 1970*; *Bentin et al., 1995*; *Röer et al., 2017a*), although in some cases task-irrelevant words, such as one's own name, may also 'break in consciousness' (*Cherry, 1953*; *Treisman, 1960*; *Wood and Cowan, 1995*; *Conway et al., 2001*).

Behavioral findings indicating linguistic processing of task-irrelevant speech have been interpreted in two opposing ways. Proponents of Late-Selection attention theories understand them as reflecting the system's capability to apply linguistic processing to more than one speech stream in parallel, albeit mostly pre-consciously (*Deutsch and Deutsch, 1963*; *Parmentier, 2008*; *Parmentier et al., 2018*; *Vachon et al., 2020*). However, others maintain an Early-Selection perspective, namely, that only one speech stream can be processed linguistically due to inherent processing bottlenecks, but that listeners may shift their attention between concurrent streams giving rise to occasional (conscious or pre-conscious) intrusions from task-irrelevant speech (*Cooke, 2006*;

*Vestergaard et al., 2011*; *Fogerty et al., 2018*). Adjudicating between these two explanations experimentally is difficult, due to the largely indirect-nature of the operationalizations used to assess linguistic processing of task-irrelevant speech. Moreover, much of the empirical evidence fueling this debate focuses on detection of individual 'task-irrelevant' words, effects that can be easily explained either by parallel processing or by attention-shifts, due to their short duration.

In attempt to broaden this conversation, here we use objective neural measures to evaluate the level of processing applied to task-irrelevant speech. Using a previously established technique of hierarchical frequency-tagging (*Ding et al., 2016*; *Makov et al., 2017*), we are able to go beyond the question of detecting individual words and probe whether linguistic processes that require integration over longer periods of time – such as syntactic structure building – are applied to task-irrelevant speech. To study this, we recorded brain activity using Magnetoencephalography (MEG) during a dichotic listening selective-attention experiment. Participants were instructed to attend to narratives of natural speech presented to one ear, and to ignore speech input from the other ear (*Figure 1*). Task-irrelevant stimuli consisted of sequences of syllables, presented at a constant rate (4 Hz), with their order manipulated to either create linguistically Structured or Non-Structured sequences. Specifically, for the Non-Structured syllables were presented in a completely random order, whereas in the Structured stimuli syllables were ordered to form coherent words, phrases, and sentences. In keeping with the frequency-tagging approach, each of these linguistic levels is associated with a different frequency (words - 2 Hz, phrases - 1 Hz, sentences - 0.5 Hz). By structuring task-irrelevant speech in this way, the two conditions were perfectly controlled for low-level acoustic attributes that contribute to energetic masking (e.g. loudness, pitch, and fine-structure), as well as for the presence of recognizable acoustic-phonetic units, which proposedly contributes to phonetic interference during speech-on-speech masking (*Rhebergen et al., 2005*; *Shinn-Cunningham, 2008*; *Kidd et al., 2016*). Rather, the only difference between the conditions was in the order of the syllables which either did or did not form linguistic structures. Consequentially, if the neural signal shows peaks at frequencies associated with linguistic-features of Structured task-irrelevant speech, as has been reported previously when these type of stimuli are attended or presented without competition (*Ding et al., 2016*; *Ding et al., 2018*; *Makov et al., 2017*), this would provide evidence that integration-based processes operating on longer time-scales are applied to task-irrelevant speech, for identifying longer linguistic units comprised of several syllables. In addition, we also tested whether the neural encoding of the to-be-attended speech itself was affected by the linguistic structure of task-irrelevant speech, which could highlight the source of potential tradeoffs or competition for resources when presented with competing speech (*Zion Golumbic et al., 2013b*; *O'Sullivan et al., 2015*; *Fiedler et al., 2019*; *Teoh and Lalor, 2019*).

## Materials and methods

### Participants

We measured MEG recordings from 30 (18 females, 12 males) native Hebrew speakers. Participants were adult volunteers, ages ranging between 18 and 34 (M = 24.8, SD = ± 4.2), and all were right-handed. Sample size was determined a priori, based on a previous study from our group using a similar paradigm and electrophysiological measures (*Makov et al., 2017*), where significant effects were found in a sample of n = 21 participants. Exclusion criteria for participation included: non-native Hebrew speakers, a history of neurological disorders or ADHD (based on self-report) or the existence of metal implants (which would disrupt MEG recordings). The study was approved by the IRB committee at Bar-Ilan University and all participants provided their written consent for participation prior to the experiment.

### Natural speech (to-be-attended)

Natural speech stimuli were narratives from publicly available Hebrew podcasts and short audio stories (duration: 44.53 ± 3.23 s). These speech materials were chosen from an existing database in the lab, that were used in previous studies and for which the behavioral task had already been validated (see Experimental Procedure). The stimuli originally consisted of narratives in both female and male voices. However, since it is known that selective attention to speech is highly influenced by whether the competing voices are of the same/different sex (*Brungart et al., 2001*; *Rivenez et al., 2006*;

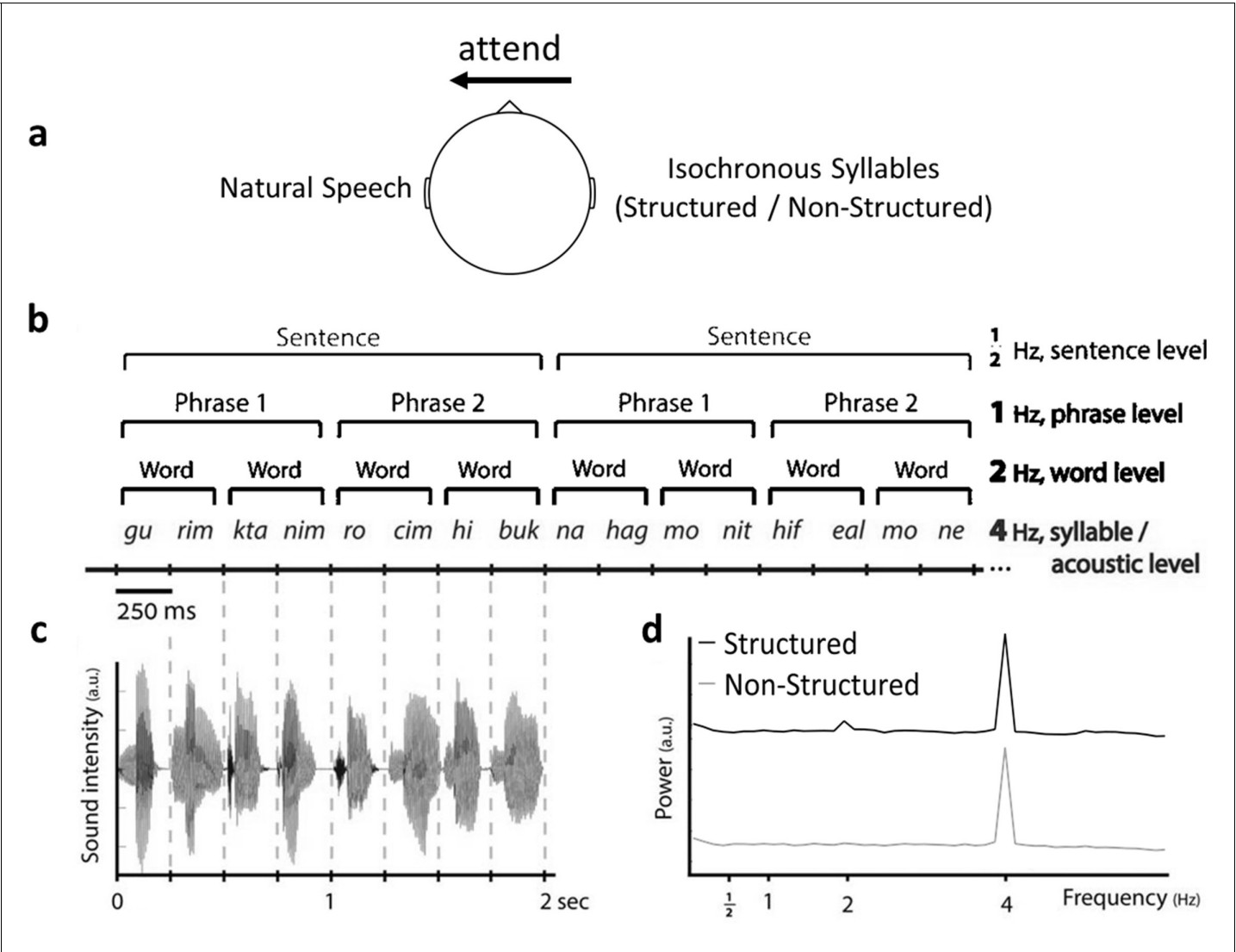

**Figure 1.** Illustration of the Dichotic Listening Paradigm. (**a**) Participants were instructed to attended right or left ear (counterbalanced) and ignore the other. To-be-attended stimulus was always natural Hebrew speech, and multiple choice questions about the content were asked at the end of each trial. The task-irrelevant ear was always presented with hierarchical frequency-tagging stimuli in two conditions: Structured and Non-Structured. (**b**) Example of intelligible (Structured) speech composed of 250 ms syllables in which 4 levels of information are differentiated based on their rate: acoustic/syllabic, word, phrasal, and sentential rates (at 4, 2, 1, and 0.5 Hz, respectively). Translation of the two Hebrew sentences in the example: 'Small puppies want a hug' and 'A taxi driver turned on the meter.' Control stimuli were Non-Structured syllable sequences with the same syllabic-rate of 4 Hz. (**c**) Representative sound wave of a single sentence (2 s). Sound intensity fluctuates at the rate of 4 Hz. (**d**) Modulation spectrua of the speech envelopes in each condition. Panels b and c are reproduced from Figure 1A and Figure 1B of *Makov et al., 2017*. Panel d has been adapted from Figure 1C of *Makov et al., 2017*.

The online version of this article includes the following figure supplement(s) for figure 1:

**Figure supplement 1.** Two examples of start and end syllables for bi-syllabic Hebrew words.

**Figure supplement 2.** The effects of syllable-order on the modulation spectrum.

*Ding and Simon, 2012*), and since the task-irrelevant stimuli were recorded only in a male voice (see below), we transformed narratives that were originally recorded in a female voice to a male voice (change-gender function in Praat; *Boersma, 2011*, http://www.praat.org). To ensure that the gender change did not affect the naturalness of the speech and to check for abnormalities in the materials, we conducted a short survey among 10 native Hebrew speakers. They all agreed that the speech sounded natural and normal. Sound intensity was equated across all narratives. Stimuli examples are available at: https://osf.io/e93qa. These natural speech narratives served as the to-be-attended

stimuli in the experiment. For each participant, they were randomly paired with task-irrelevant speech (regardless of condition), to avoid material-specific effects.

## Frequency-tagged speech (task-irrelevant)

A bank of individually recorded Hebrew syllables were used to create two sets of isochronous speech sequences. Single syllables were recorded in random order by a male actor, and remaining prosodic cues were removed using pitch normalization in Praat. Additional sound editing was performed to adjust the length of each syllable to be precisely 250 ms either by truncation or silence padding at the end (original mean duration 243.6 ± 64.3 ms, range 168–397 ms). In case of truncation, a fading out effect was applied to the last 25 ms to avoid clicks. Sound intensity was then manually equated for all syllables.

These syllables were concatenated to create long sequences using custom-written scripts in MATLAB (The MathWorks; code available at https://osf.io/e93qa), equated in length to those of the natural speech segments (44.53 ± 3.23 seconds). Sequences could either be Non-Structured, with syllables presented in a fully random order without creating meaningful linguistic units, or they could be linguistically Structured. Structured sequences were identical to those used in a previous study from our group (*Makov et al., 2017*), and were formed as follows: Every two syllables formed a word, every two words formed a phrase, and every two phrases formed a sentence. Because syllables were grouped hierarchically into linguistic constituents with no additional acoustic gaps inserted between them, different linguistic hierarchies are associated with fixed periodicities throughout the stimuli (syllables at 4 Hz, words at 2 Hz, phrases at 1 Hz, and sentences at 0.5 Hz; *Figure 1b*). Structured stimuli also contained no prosodic cues or other low-level acoustic indications for boundaries between linguistics structures, nor did Structured sentences include rhymes, passive form of verbs, or arousing semantic content. See Supplementary Material for more information on the construction of Structured and Non-Structured stimuli.

The modulation spectrum of both types of task-irrelevant stimuli is shown in *Figure 1d*. It was calculated using a procedure analogous to the spectral analysis performed on the MEG data, in order to ensure maximal comparability between the spectrum of the stimuli and the spectrum of the neural response. Specifically, (1) the broadband envelope of each sequence was extracted by taking the root-mean-square of the audio (10 ms smoothing window); (2) the envelope was segmented into 8 s long segments, which was identical to the segmentation of the MEG data; (3) a fast Fourier transform (FFT) was applied to each segment; and (4) averaged across segments. As expected, both stimuli contained a prominent peak at 4 Hz, corresponding to the syllable-rate. The Structured stimuli also contain a smaller peak at 2 Hz, which corresponds to the word-rate. This is an undesirable side-effect of the frequency-tagging approach, since ideally these stimuli should not contain any energy at frequencies other than the syllable-rate. As shown in the Supplementary Material, the 2 Hz peak in the modulation spectrum reflects the fact that a consistently different subset of syllables occurs at each position within the sequence (e.g. at the beginning\end of words; for similar acoustic effects when using frequency-tagging of multi-syllable words see *Luo and Ding, 2020*. A similar 2 Hz peak is not observed in the Non-Structured condition, where the syllables are randomly positioned throughout all stimuli. Given this difference in the modulation spectrum, if we were to observe a 2 Hz peak in the neural response to Structured vs. Non-Structured stimuli, this would not necessarily provide conclusive evidence for linguistic 'word-level' encoding (although see *Makov et al., 2017* and Supplementary Material for a way to control for this). As it happened, in the current dataset we did not see a 2 Hz peak in the neural response in either condition (see Results), therefore this caveat did not affect the interpretability of the data in this specific instance. Importantly, neither the Structured nor the Non-Structured stimuli contained peaks at frequencies corresponding to other linguistic levels (1 Hz and 0.5 Hz), hence comparison of neural responses at these frequencies remained experimentally valid. Stimuli examples are available at: https://osf.io/e93qa.

## Experimental procedure

The experiment used a dichotic listening paradigm, in which participants were instructed to attend to a narrative of natural Hebrew speech presented to one ear, and to ignore input from the other ear where either Structured or Non-Structured task-irrelevant speech was presented. The experiment included a total of 30 trials (44.53 ± 3.23 seconds), and participants were informed at the

beginning of each trial which ear to attend to, which was counterbalanced across trials. The sound intensity of the task-irrelevant stimuli increased gradually during the first three seconds of each trial, to avoid inadvertent hints regrading word-boundaries, and the start of each trial was excluded from data analysis. After each trial, participants answered four multiple choice questions about the content of the narrative they were supposed to attend to (3-answers per question; chance level = 0.33). Some of the questions required recollection of specific details (e.g. 'what color was her hat?"), and some addressed the 'gist' of the narrative, (e.g. 'why was she sad?"). The average accuracy rate of each participant (% questions answered correctly) was calculated across all questions and narratives, separately for trials in the Structured and Non-Structured condition.

This task was chosen as a way to *motivate and guide* participants to direct attention toward the to-be-attended narrative and provide *verification* that indeed they listened to it. At the same time, we recognize that this task is not highly sensitive for gauging the full extent of processing the narrative for two reasons: (1) its sparse sampling of behavior (four questions for a 45 s narrative); and (2) since accuracy is affected by additional cognitive factors besides attention such as short-term memory, engagement, and deductive reasoning. Indeed, behavioral screening of this task showed that performance was far from perfect even when participants listened to these narratives in a single-speaker context (i.e., without additional competing speech; average accuracy rate 0.83 ± 0.08; n = 10). Hence, we did not expect performance on this task to necessarily reflect participants' internal attentional state. At the same time, this task is instrumental in guiding participants' selective attention toward the designated speaker, allowing us to analyze their neural activity during uninterrupted listening to continuous speech, which was the primarily goal of the current study.

## Additional tasks

Based on previous experience, the Structured speech materials are not immediately recognizable as Hebrew speech and require some familiarization. Hence, to ensure that all participants could identify these as speech, and in order to avoid any perceptual learning effects during the main experiment, they underwent a familiarization stage prior to the start of the main experiment, inside the MEG. In this stage, participants heard sequences of 8 isochronous syllables, which were either Structured – forming a single sentence – or Non-Structured. After each trial participants were asked to repeat the sequence out loud. The familiarity stage continued until participants correctly repeated five stimuli of each type. Structured and Non-Structured stimuli were presented in random order.

At the end of the experiment, a short auditory localizer task was performed. The localizer included hearing tones in five different frequencies: 400, 550, 700, 850, and 1000 Hz, all 200 ms long. The tones were presented with random ISIs: 500, 700, 1000, 1200, 1400, 1600, 1800, and 2000 ms. Participants listened to the tones passively and were instructed only to focus on the fixation mark in the center of the screen.

## MEG data acquisition

MEG recordings were conducted with a whole-head, 248-channel magnetometer array (4D Neuroimaging, Magnes 3600 WH) in a magnetically shielded room at the Electromagnetic Brain Imaging Unit, Bar-Ilan University. A series of magnetometer and gradiometer reference coils located above the signal coils were used to record and subtract environmental noise. The location of the head with respect to the sensors was determined by measuring the magnetic field produced by small currents delivered to five head coils attached to the scalp. Before the experimental session, the position of head coils was digitized in relation to three anatomical landmarks (left and right preauricular points and nasion). The data was acquired at a sample rate of 1017.3 Hz and an online 0.1 to 200 Hz band-pass filter was applied. The 50 Hz power line noise fluctuations were recorded directly from the power line as well as vibrations using a set of accelerometers attached to the sensor in order to remove the artifacts on the MEG recordings.

## MEG preprocessing

Preprocessing was performed in MATLAB (The MathWorks) using the FieldTrip toolbox (http://www.fieldtriptoolbox.org). Outlier trials were identified manually by visual inspection and were excluded from analysis. Using independent component analysis (ICA) we removed eye movements (EOG), heartbeat and vibrations of the building. The clean data was then segmented into 8 s long

segments, which corresponds to four sentences in the Structured condition. Critically, these segments were perfectly aligned such that they all start with the onset of a syllable, which in the Structured condition will also be the onset of a sentence.

## Source estimation

Source estimation was performed in Python (http://www.python.org) using the MNE-python platform (*Gramfort et al., 2013*; *Gramfort et al., 2014*). Source modeling was performed on the preprocessed MEG data, by computing Minimum-Norm Estimates (MNEs). In order to calculate the forward solution, and constrain source locations to the cortical surface, we constructed a Boundary Element Model (BEM) for each participant. BEM was calculated using the participants' head shape and location relative to the MEG sensors, which was co-registered to an MRI template (FreeSurfer; surfer.nmr.mgh.harvard.edu). Then, the cortical surface of each participant was decimated to 8194 source locations per hemisphere with at least 5 mm spacing between adjacent locations. A noise covariance matrix was estimated using the inter-trial intervals in the localizer task (see Additional Tasks), that is, periods when no auditory stimuli were presented. Then, an inverse operator was computed based on the forward solution and the noise covariance matrix, and was used to estimate the activity at each source location. For visualizing the current estimates on the cortical surface, we used dynamic Statistical Parametric Map (dSPM), which is an F-statistic calculated at each voxel and indicating the relationship between MNE amplitude estimations and the noise covariance (*Dale et al., 2000*). Finally, individual cortical surfaces were morphed onto a common brain, with 10,242 dipoles per hemisphere (*Fischl et al., 1999*), in order to compensate for inter-subject differences.

## Behavioral data analysis

The behavioral score was calculated as the average correct response across trials (four multiple-choice question per narrative) for each participant. In order to verify that participants understood and completed the task, we performed a t-test between accuracy rates compared to chance-level (i.e. 0.33).Then, to test whether performance was affected by the type of task-irrelevant speech presented, we performed a paired t-test between the accuracy rates in both conditions. We additionally performed a median-split analysis of the behavioral scores across participant, based on their neural response to task-irrelevant speech (specifically the phrase-level response; see MEG data analysis), to test for possible interactions between performance on the to-be-attended speech and linguistic neural representation of task-irrelevant speech.

## MEG data analysis

### Spectral analysis

#### Scalp-level

Inter-trial phase coherence (ITPC) was calculated on the clean and segmented data. To this end, we applied a Fast Fourier Transform (FFT) to individual 8 s long segments and extracted the phase component at each frequency (from 0.1 to 15 Hz, with a 0.125 Hz step). The normalized (z-scored) ITPC at each sensor was calculated using the Matlab circ_rtest function (circular statistics toolbox; *Berens, 2009*). This was performed separately for the Structured and Non-Structured conditions. In order to determine which frequencies had significant ITPC, we performed a t-test between each frequency bin relative to the surrounding frequencies (average two bins from each side), separately for each condition (*Nozaradan et al., 2018*). In addition, we directly compared the ITPC spectra between the two conditions using a permutation test. In each permutation, the labels of the two conditions were randomly switched in half of the participants, and a paired t-test was performed. This was repeated 1000 times, creating a null-distribution of t-values for this paired comparison, separately for each of the frequencies of interest. The t-values of the real comparisons were evaluated relative to this null-distribution, and were considered significant if they fell within the top 5% (one-way comparison, given our a-priori prediction that peaks in the Structured condition would be higher than in the Non-Structured condition). This procedure was performed on the average ITPC across all sensors, to avoid the need to correct for multiple comparisons, and focused specifically on four frequencies of interest (FOI): 4, 2, 1, and 0.5 Hz which correspond to the four linguistic levels present in the Structured stimuli (syllables, words, phrases and sentences, respectively).

## Source-level

Spectral analysis of source-level data was similar to that performed at the sensor level. ITPC was calculated for each frequency between 0.1 and 15 Hz (0.125 Hz step) and a t-test between the ITPC at each frequency relative to the surrounding frequencies (two bins from each side) was performed in order to validate the response peaks at the FOIs. Then, statistical comparison of responses at the source-level in the Structured and Non-Structured conditions focused only on the peaks that showed a significant difference between conditions at the scalp-level (in this case, the peak at 1 Hz). In order to determine which brain-regions contributed to this effect, we used 22 pre-defined ROIs in each hemisphere, identified based on multi-modal anatomical and functional parcellation (*Glasser et al., 2016*; Supplementary Neuroanatomical Results [table 1, page 180] and see Figure 3c). We calculated the mean ITPC value in each ROI, across participants and conditions, and tested for significant differences between them using a permutation test, which also corrected for multiple comparisons. As in the scalp-data, the permutation test was based on randomly switching the labels between conditions for half of the participants and conducting a paired t-test within each ROI. In each permutation we identified ROIs that passed a statistical threshold for a paired t-test ($p<0.05$) and the sum of their t-values was used as a global statistic. This procedure was repeated 1000 times, creating a null-distribution for this global statistic. A similar procedure was applied to the real data, and if the global statistic (sum of t-values in the ROIs that passed an uncorrected threshold of $p<0.05$) fell within the top 5% of the null-distribution, the entire pattern could be considered statistically significant. This procedure was conducted separately within each hemisphere.

## Speech-tracking analysis

Speech-tracking analysis was performed in order to estimate the neural response to the natural speech that served as the to-be-attended stimulus. To this end, we estimated the Temporal Response Function (TRF), which is a linear transfer-function expressing the relationship between features of the presented speech stimulus $s(t)$ and the recorded neural response $r(t)$. TRFs were estimated using normalized reverse correlation as implemented in the STRFpak Matlab toolbox (strfpak. berkeley.edu) and adapted for MEG data (*Zion Golumbic et al., 2013a*). Tolerance and sparseness factors were determined using a jackknife cross-validation procedure, to minimize effects of over-fitting. In this procedure, given a total of $N$ trials, a TRF is estimated between $s(t)$ and $r(t)$ derived from N-1 trials, and this estimate is used to predict the neural response to the left-out stimulus. The tolerance and sparseness factors that best predicted the actual recorded neural signal (predictive power estimated using Pearson's correlation) were selected based on scalp-level TRF analysis (collapsed across conditions), and these were also used when repeating the analysis on the source-level data (*David et al., 2007*). The predictive power of the TRF model was also evaluated statistically by comparing it to a null-distribution obtained from repeating the procedure on 1000 permutations of mismatched $s^*(t)$ and $r^*(t)$.

TRFs to the to-be-attended natural speech were estimated separately for trials in which the task-irrelevant speech was Structured vs. Non-Structured. TRFs in these two conditions were then compared statistically to evaluate the effect of the type of task-irrelevant stimulus on neural encoding of to-be-attended speech. For scalp-level TRFs, we used a spatial-temporal clustering permutation test to identify the time-windows where the TRFs differed significantly between conditions (fieldtrip toolbox; first level stat $p<0.05$, cluster corrected). We then turned to the source-level TRFs to further test which brain regions showed significant difference between conditions, by estimating TRFs in the same 22 pre-defined ROIs in each hemisphere used above. TRFs in the Structured vs. Non-Structured conditions were compared using t-tests in 20 ms long windows (focusing only on the 70–180 ms time window which was found to be significant in the scalp-level analysis) and corrected for multiple comparisons using spatial-temporal clustering permutation test.

## Results

Data from one participant was excluded from all analyses due to technical problems during MEG recording. Six additional participants were removed only from source estimation analysis due to technical issues. The full behavioral and neural data are available at: https://osf.io/e93qa.

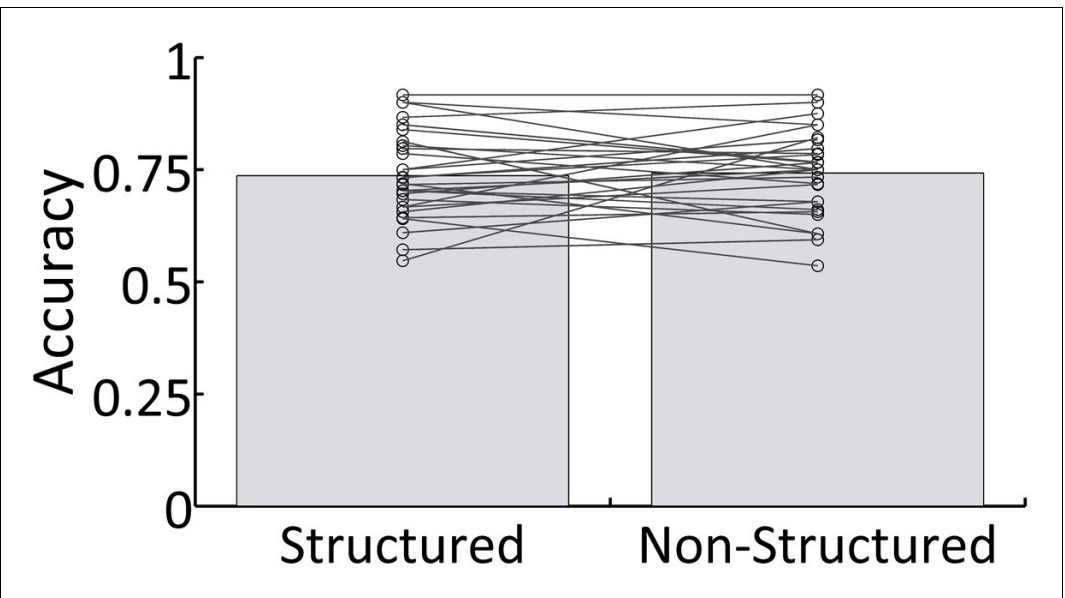

**Figure 2.** Behavioral results. Mean accuracy across all participants for both Structured and Non-Structured conditions. Lines represent individual results.

## Behavioral results

Behavioral results reflecting participants response accuracy on comprehension questions about narratives were significantly above chance (M = 0.715, SD = ± 0.15; t(28)=26.67, p<0.001). There were no significant differences in behavior as a function of whether task-irrelevant speech was Structured or Non-Structured (t(28)=−0.31, p=0.75; *Figure 2*). Additionally, to test for possible interactions between answering questions about the to-be-attended speech and linguistic neural representation of task-irrelevant speech, we performed a median-split analysis of the behavioral scores across participants. Specifically, we used the magnitude of the ITPC value at 1 Hz in the Structured condition (averaged across all sensors), in order to split the sample into two groups – with high and low 1 Hz responses. We performed a between-group t-test on the behavioral results in the Structured condition, and also on the difference between conditions (Structured – Non-Structured). Neither test showed significant differences in performance between participants whose 1 Hz ITPC was above vs. below the median (Structured condition: t(27) = −1.07, p=0.29; Structured – Non-Structured: t(27) = −1.04, p=0.15). Similar null-results were obtained when the median-split was based on the source-level data.

## Hierarchical frequency-tagged responses to task-irrelevant speech

Scalp-level spectra of the Inter-trial phase coherence (ITPC) showed a significant peak at the syllabic-rate (4 Hz) in response to both Structured and Non-Structured hierarchical frequency-tagged speech, with a four-pole scalp-distribution common to MEG recorded auditory responses (*Figure 3a*) (p<10⁻9; large effect size, Cohen's d > 1.5 in both). As expected, there was no significant difference between Structured and Non-Structured condition in the 4 Hz response (p=0.899). Importantly, we also observed a significant peak at 1 Hz in the Structured condition (p<0.003; moderate effect size, Cohen's d = 0.6), but not in the Non-Structured condition (p=0.88). Comparison of the 1 Hz ITPC between these conditions also confirmed a significant difference between them (p=0.045; moderate effect size, Cohen's d = 0.57). The scalp-distribution of the 1 Hz peak did not conform to the typical auditory response topography, suggesting different neural generators. No other significant peaks were observed at any other frequencies, including the 2 Hz or 0.5 Hz word and sentence-level rates, nor did the responses at these rates differ significantly between conditions.

In order to determine the neural source of the 1 Hz peak in the Structured condition, we repeated the spectral analysis in source-space. An inverse solution was applied to individual trials and the ITPC was calculated in each voxel. As shown in *Figure 3b*, the source-level ITPC spectra, averaged

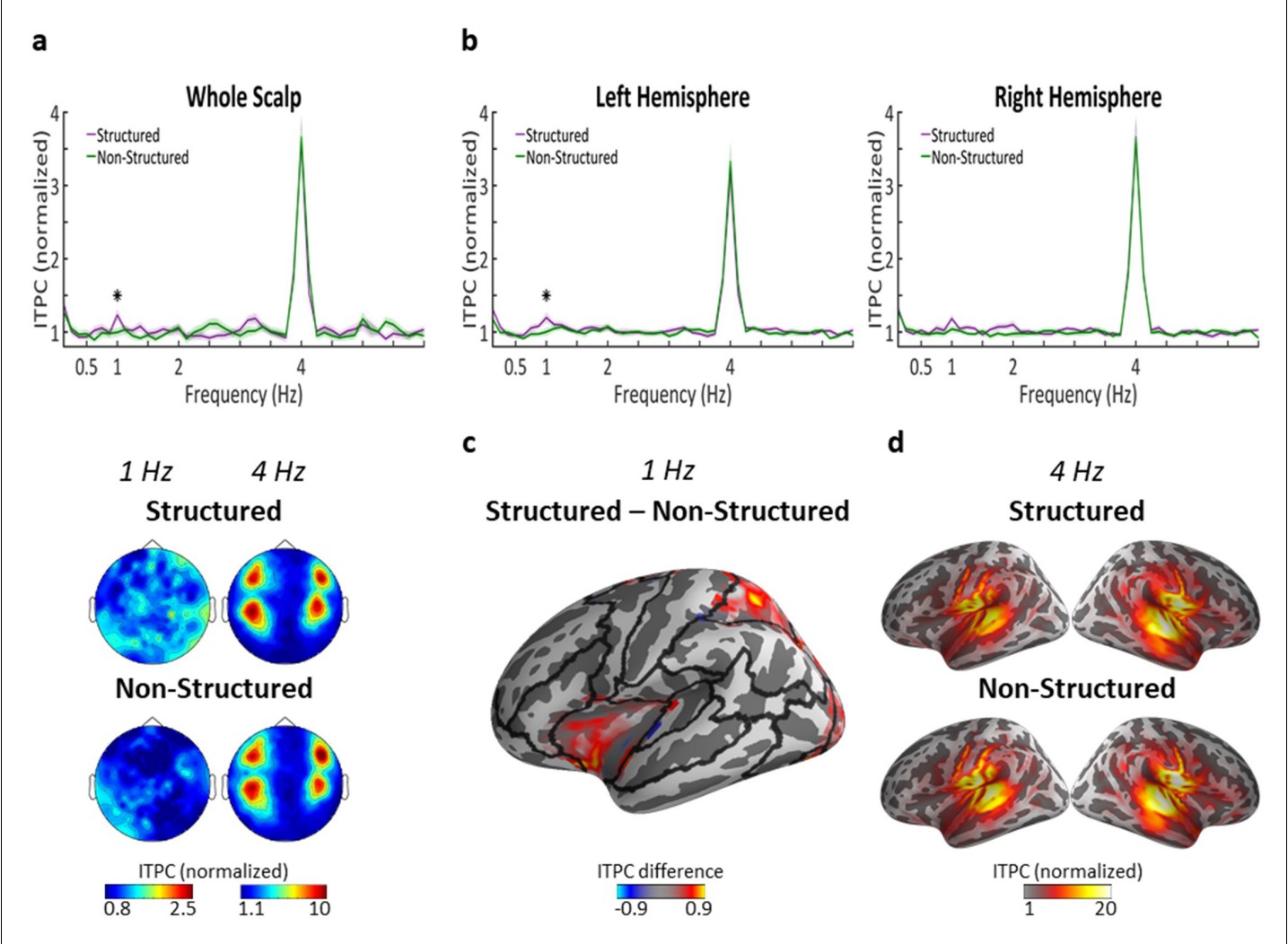

**Figure 3.** Neural tracking of linguistic structures in task-irrelevant speech. (**a**) Top panel shows the ITPC spectrum at the scalp-level (average across all sensors) in response to Structured (purple) and Non-Structured (green) task-irrelevant speech. ITPC values z-score normalized, as implemented in the circ_rtest function (see Materials and methods). Shaded areas indicate SEM across participants (n = 29). Asterisk represents statistically significant difference (p<0.05) between conditions, indicating a significant response at 1 Hz for Structured task-irrelevant speech, which corresponds to the phrase-level. Bottom panel shows the scalp-topography of ITPC at 4 Hz and 1 Hz in the two conditions. (**b**) ITPC spectrum at the source-level, averaged across all voxels in each hemisphere. Shaded highlights denote SEM across participants (n = 23). Asterisk represents statistically significant difference (p<0.05) between conditions, indicating a significant response at 1 Hz for Structured task-irrelevant speech in the left, but not right hemisphere. (**c**) Source-level map on a central inflated brain depicting the ROIs in the left hemisphere where significant differences in ITPC at 1 Hz were found for Structured vs. Non-Structured task-irrelevant speech. Black lines indicate the parcellation into the 22 ROIs used for source-level analysis. (**d**) Source-level maps showing localization of the syllabic-rate response (4 Hz) in both conditions.

over each hemisphere separately, is qualitatively similar to that observed at the scalp-level. The only significant peaks were at 4 Hz in both conditions (p<10$^{-8}$; large effect size, Cohen's d > 1.7 in both conditions and both hemispheres) and at 1 Hz in the Structured condition (left hemisphere p=0.052, Cohen's d = 0.43; right hemisphere p<0.007, Cohen's d = 0.6), but not in the Non-Structured condition. Statistical comparison of the 1 Hz peak between conditions revealed a significant difference between the Structured and Non-Structured condition over the left hemisphere (p=0.026, Cohen's d = 0.57), but not over the right hemisphere (p=0.278).

*Figure 3c* shows the source-level distribution within the left hemisphere of the difference in 1 Hz ITPC between the Structured and Non-Structured condition. The effect was observed primarily in frontal and parietal regions. Statistical testing evaluating the difference between conditions was performed in 22 pre-determined ROIs per hemisphere, using a permutation test. This indicated

significant effects in several ROIs in the left hemisphere including the inferior-frontal cortex and superior parietal cortex, as well as the mid-cingulate and portions of the middle and superior occipital gyrus (cluster-corrected p=0.002). No ROIs survived multiple-comparison correction in the right hemisphere (cluster-corrected p=0.132), although some ROIs in the right cingulate were significant at an uncorrected level (p<0.05).

With regard to the 4 Hz peak, it was localized as expected to bilateral auditory cortex and did not differ significantly across conditions in either hemisphere (left hemisphere: p=0.155, right hemisphere: p=0.346). We therefore did not conduct a more fine-grained analysis of different ROIs. As in the scalp-level data, no peaks were observed at 2 Hz and no significant difference between conditions (left hemisphere: p=0.963, right hemisphere: p=0.755).

### Speech tracking of to-be-attended speech

Speech tracking analysis of responses to the to-be-attended narrative yielded robust TRFs (scalp-level predictive power r = 0.1, p<0.01 vs. permutations). The TRF time-course featured two main peaks, one ~ 80 ms and the other ~140 ms, in line with previous TRF estimations (*Akram et al., 2017*; *Fiedler et al., 2019*; *Brodbeck et al., 2020b*). Both the scalp-level and source-level TRF analysis indicated that TRFs were predominantly auditory – showing the common four-pole distribution at the scalp-level (*Figure 4a*) and in the source-level analysis was localized primarily to auditory cortex (superior temporal gyrus and sulcus; STG/STS) as well as left insula/IFG (*Figure 4b*; 140 ms).

When comparing the TRFs to the to-be-attended speech as a function of whether the competing task-irrelevant stimulus was Structured vs. Non-Structured, some interesting differences emerged. Spatial-temporal clustering permutation test on the scalp-level TRFs revealed significant differences between the conditions between 70–180 ms (p<0.05; cluster corrected), including both the early

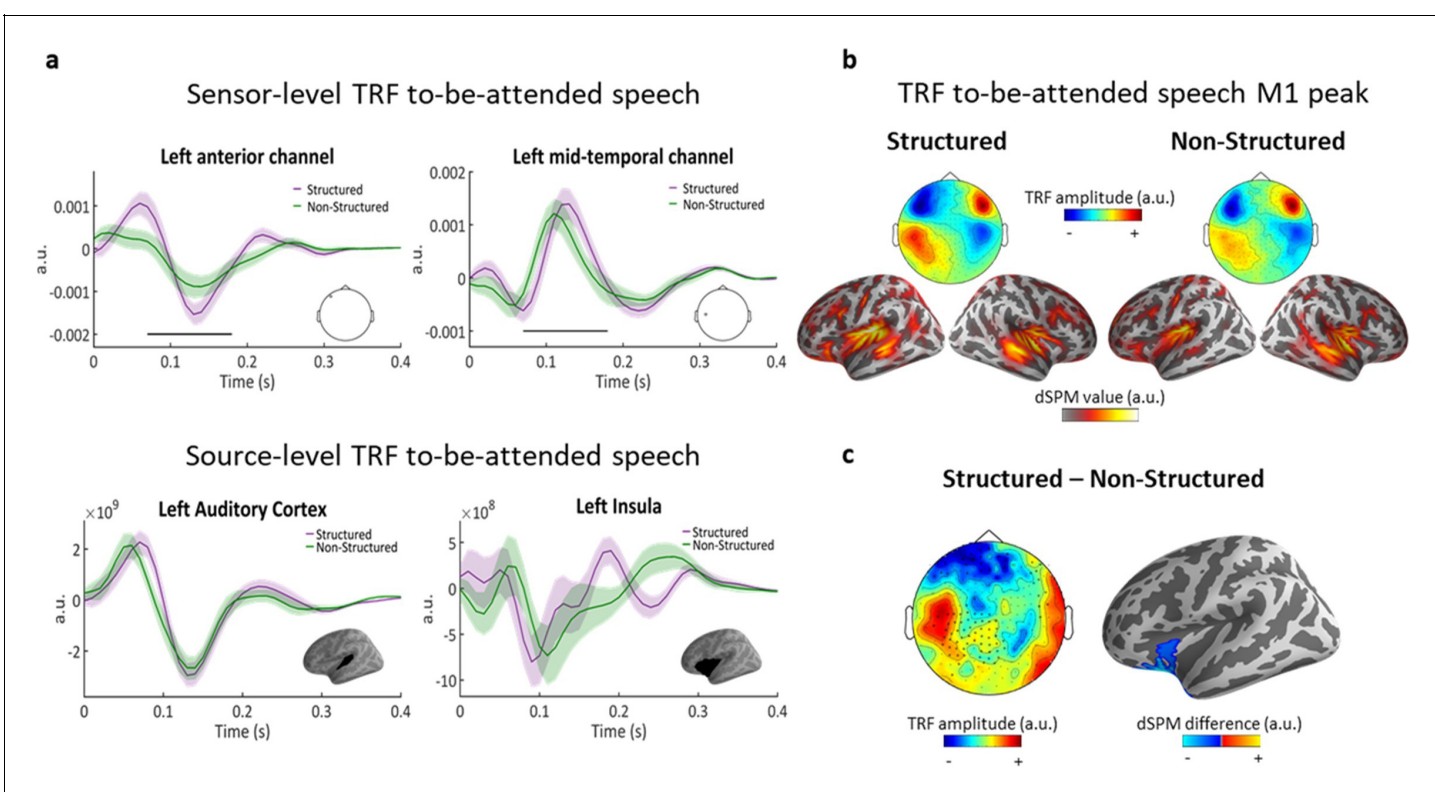

**Figure 4.** Speech tracking of to-be-attended speech. (a) top, TRF examples from two sensors, showing the positive and negative poles of the TRF over the left hemisphere. Shaded highlights denote SEM across participants. Black line indicates the time points where the difference between conditions was significant (spatio-temporal cluster corrected). bottom, TRF examples from source-level ROIs in left Auditory Cortex and in the left Inferior Frontal/Insula region. (b) topographies (top) and source estimations (bottom) for each condition at M1 peak (140 ms). (c) left, topography of the difference between conditions (Structured – Non-Structured) at the M1 peak (140 ms). Asterisks indicate the MEG channels where this difference was significant (cluster corrected). right, significant cluster at the source-level.

and late TRF peaks, at a large number of sensors primarily on the left (*Figure 4a and c*). Specifically, TRF responses to the to-be-attended speech were enhanced when the task-irrelevant stimulus was Structured vs. Non-Structured. The effect was observed at the scalp-level with opposite polarity in frontal vs. medial sensors, and was localized at the source-level to a single cluster in the left inferior-frontal cortex, that included portions of the insula and orbito-frontal cortex (*Figure 4c*; spatial-temporal clustering p=0.0058).

## Discussion

In this MEG study, frequency-tagged hierarchical speech was used to probe the degree to which linguistic processing is applied to task-irrelevant speech, and how this interacts with processing speech that is presumably in the focus of attention ('to-be-attended'). As expected, we observe obligatory acoustic representation of task-irrelevant speech, regardless of whether it was Structured or Non-Structured, which manifest as a 4 Hz syllable-level response localized to bilateral auditory regions in the STG/STS. Critically, for Structured task-irrelevant speech, we also find evidence for neural tracking of the phrasal structure, with a 1 Hz peak localized primarily to left inferior-frontal cortex and left posterior-parietal cortex. The regions are not associated with low-level processing, but rather play an important role in speech processing and higher order executive functions (*Dronkers et al., 2004*; *Humphries et al., 2006*; *Linden, 2007*; *Corbetta et al., 2008*; *Edin et al., 2009*; *Hahn et al., 2018*). Additionally, we find that the speech-tracking response to the *to-be-attended* speech in left inferior frontal cortex was also affected by whether task-irrelevant speech was linguistically Structured or not. These results contribute to ongoing debates regarding the nature of the competition for processing resources during speech-on-speech masking, demonstrating that linguistic processes requiring integration of input over relatively long timescales can indeed be applied to task-irrelevant speech.

### The debate surrounding linguistic processing of task-irrelevant speech

Top-down attention is an extremely effective process, by which the perceptual and neural representation of task-relevant speech are enhanced at the expense of task-irrelevant stimuli, and speech in particular (*Horton et al., 2013*; *Zion Golumbic et al., 2013b*; *O'Sullivan et al., 2015*; *Fiedler et al., 2019*; *Teoh and Lalor, 2019*). However, the question still stands: what degree of linguistic processing is applied to task-irrelevant speech? One prominent position is that attention is required for linguistic processing and therefore speech that is outside the focus of attention is not processed beyond its sensory attributes (*Lachter et al., 2004*; *Brodbeck et al., 2018a*; *Ding et al., 2018*). However, several lines of evidence suggest that linguistic features of task-irrelevant speech can be processed as well, at least under certain circumstances. For example, task-irrelevant speech is more disruptive to task performance if it is intelligible, as compared to unintelligible noise-vocoded or rotated speech (*Marrone et al., 2008*; *Iyer et al., 2010*; *Best et al., 2012*; *Gallun and Diedesch, 2013*; *Swaminathan et al., 2015*; *Kidd et al., 2016*) or a foreign language (*Freyman et al., 2001*; *Rhebergen et al., 2005*; *Cooke et al., 2008*; *Calandruccio et al., 2010*; *Francart et al., 2011*). This effect, referred to as informational masking, is often attributed to the detection of familiar acoustic-phonetic features in task-irrelevant speech, that can lead to competition for phonological processing ('phonological interference') (*Durlach et al., 2003*; *Drullman and Bronkhorst, 2004*; *Kidd et al., 2008*; *Shinn-Cunningham, 2008*; *Rosen et al., 2013*). However, the phenomenon of informational masking alone is insufficient for determining the extent to which task-irrelevant speech is processed beyond identification of phonological units.

Other lines of investigation have focused more directly on whether task-irrelevant speech is represented at the semantic level. Findings that individual words from a task-irrelevant source are occasionally detected and recalled, such as one's own name, (*Cherry, 1953*; *Wood and Cowan, 1995*; *Conway et al., 2001*; *Röer et al., 2017b*; *Röer et al., 2017a*), have been taken as evidence that task-irrelevant inputs can be semantically processed, albeit the information may not be consciously available. Along similar lines, a wealth of studies demonstrate the 'Irrelevant Sound Effect' (ISE), showing that the semantic content of task-irrelevant input affects performance on a main task, mainly through priming effects and interference with short-term memory (*Lewis, 1970*; *Bentin et al., 1995*; *Surprenant et al., 1999*; *Dupoux et al., 2003*; *Beaman, 2004*; *Rivenez et al., 2006*; *Beaman et al., 2007*; *Rämä et al., 2012*; *Aydelott et al., 2015*; *Schepman et al., 2016*;

*Vachon et al., 2020*). However, an important caveat precludes interpreting these findings as clear-cut evidence for semantic processing of task-irrelevant speech: Since these studies primarily involve presentation of arbitrary lists of words (mostly nouns), usually at a relatively slow rate, an alternative explanation is that the ISE is simply a result of occasional shifts of attention toward task-irrelevant stimuli (*Carlyon, 2004*; *Lachter et al., 2004*). Similarly, the effects of informational masking discussed above can also be attributed to a similar notion of perceptual glimpsing, that is gleaning bits of the task-irrelevant speech in the short 'gaps' in the speech that is to-be-attended (*Cooke, 2006*; *Kidd et al., 2016*; *Fogerty et al., 2018*). These claims – that effects of task-irrelevant speech are not due to parallel processing but reflect shifts of attention – are extremely difficult to reject empirically, as they would require insight into the listeners' internal state of attention, which at present is not easy to operationalize.

## Phrase-level response to task-irrelevant speech

In attempt to broach the larger question of processing task-irrelevant speech, the current study takes a different approach by focusing not on detection of single words, but on linguistic processes that operate over longer timescales. To this end the stimuli used here, in both the to-be-attended and the task-irrelevant ear, was continuous speech rather than word-lists whose processing requires accumulating and integrating information over time, which is strikingly different than the point-process nature of listening to word-lists (*Fedorenko et al., 2016*). Using continuous speech is also more representative of the type of stimuli encountered naturally in the real world (*Hill and Miller, 2010*; *Risko et al., 2016*; *Matusz et al., 2019*; *Shavit-Cohen and Zion Golumbic, 2019*). In addition, by employing hierarchical frequency-tagging, we were able to obtain objective and direct indications of which levels of information were detected within task-irrelevant speech. Indeed, using this approach we were able to identify a phrase-level response for Structured task-irrelevant speech, which serves as a positive indication that these stimuli are indeed processed in a manner sufficient for identifying the boundaries of syntactic structures.

An important question to ask is whether the phrase-level response observed for task-irrelevant speech can be explained by attention shifts? Admittedly, in the current design participants could shift their attention between streams in an uncontrolled fashion, allowing them to 'glimpse' portions of the task-irrelevant speech, integrate and comprehend (portions of) it. Indeed, this is one of the reasons we refrain from referring to the task-irrelevant stream as 'unattended': since we have no principled way to empirically observe the internal loci or spread of attention, we chose to focus on its behavioral relevance rather than make assumptions regarding the participants' attentional-state. Despite the inherent ambiguity regarding the underlying dynamics of attention, the fact that here we observe a phrase-level response for task-irrelevant speech is direct indication that phonetic-acoustic information from this stream was decoded and integrated over time, allowing the formation of long-scale representations for phrasal boundaries. If this is a result of internal 'glimpsing', this would imply either that (a) the underlying hierarchical speech structure was detected and used to guide 'glimpses' in a rhythmic-manner to points in time that are most informative; or (b) that 'glimpses' occur irregularly, but that sufficient information is gleaned through them and stored in working-memory to allow the consistent detection phrasal boundaries in task-irrelevant speech. Both of these options imply a sophisticated multiplexed encoding-scheme for successful processing of concurrent speech, that relies on precise temporal control and working-memory storage. Another possibility, of course, is that there is no need for attention-shifts and that the system has sufficient capacity to process task-irrelevant speech in parallel to focusing primarily on the to-be-attended stream. As mentioned above, the current data cannot provide insight into which of these listening-strategies underlies the generation of the observed phrase-level response to task-irrelevant speech. However, we hope that future studies will gain empirical access into the dynamic of listeners' internal attentional state and help shed light on this pivotal issue.

The current study is similar in design to another recent study by *Ding et al., 2018* where Structured frequency-tagged speech was presented as a task-irrelevant stimulus. In contrast to the results reported here, they did not find significant peaks at any linguistic-related frequencies in the neural response to task-irrelevant speech. In attempt to resolve this discrepancy, it is important to note that these two studies differ in an important way – in the listening effort that was required of participants in order to understand the to-be-attended speech. While in the current experiment to-be-attended speech was presented in its natural form, mimicking the listening effort of real-life speech-

processing, in the study by *Ding et al., 2018* to-be-attended speech was time-compressed by a factor of 2.5 and naturally occurring gaps were removed, making the comprehension task substantially more effortful (*Nourski et al., 2009*; *Müller et al., 2019*). Load Theory of Attention proposes that the allocation of processing resources among competing inputs can vary as a function of the perceptual traits and cognitive load imposed by the task (*Lavie et al., 2004*; *Murphy et al., 2017*). Accordingly, it is plausible that these divergent results are due to the extreme difference in the perceptual load and listening effort in the two studies. Specifically, if understanding the to-be-attended speech imposes relatively low perceptual and cognitive load, then sufficient resources may be available to additionally process aspects of task-irrelevant speech, but that this might not be the case as the task becomes more difficult and perceptually demanding (*Wild et al., 2012*; *Gagné et al., 2017*; *Peelle, 2018*).

More broadly, the comparison between these two studies invites re-framing of the question regarding the type/level of linguistic processing applied to task-irrelevant speech, and propels us to think about this issue not as a yes-or-no dichotomy, but perhaps as a more flexible process that depends on the specific context (*Brodbeck et al., 2020b*). The current results provide a non-trivial positive example for processing task-irrelevant speech that is indeed processed beyond its acoustic attributes, in an experimental context that closely emulates the perceptual and cognitive load encountered in real-life (despite the admitted unnatural nature of the task-irrelevant speech). At the same time, they do not imply that this is always the case, as is evident from the diverse results reported in the literature regarding processing task-irrelevant speech, as discussed at length above. Rather, they invite adopting a more flexible perspective of processing bottlenecks within the speech processing system, that takes into consideration the perceptual and cognitive load imposed in a given context, in line with load theory of attention (*Mattys et al., 2012*; *Lavie et al., 2014*; *Fairnie et al., 2016*; *Gagné et al., 2017*; *Peelle, 2018*). Supporting this perspective, others have also observed that the level of processing applied to task-irrelevant stimuli can be affected by task demands (*Hohlfeld and Sommer, 2005*; *Pulvermüller et al., 2008*). Moreover, individual differences in attentional abilities, and particularly the ability to process concurrent speech, have been attributed partially to working-memory capacity, a trait associated with the availability of more cognitive resources (*Beaman et al., 2007*; *Forster and Lavie, 2008*; *Naveh-Benjamin et al., 2014*; *Lambez et al., 2020*) but cf. (*Elliott and Briganti, 2012*). As cognitive neuroscience research increasingly moves toward studying speech processing and attention in real-life circumstances, a critical challenge will be to systematically map out the perceptual and cognitive factors that contribute to, or hinder, the ability to glean meaningful information from stimuli that are outside the primary focus of attention.

## The brain regions where phrase-level response is observed

The phrase-level neural response to task-irrelevant Structured speech was localized primarily to two left-lateralized clusters: one in the left anterior fronto-temporal cortex and the other in left posterior-parietal cortex. The fronto-temporal cluster, which included the IFG and insula, is known to play an important role in speech processing (*Dronkers et al., 2004*; *Humphries et al., 2006*; *Brodbeck et al., 2018b*; *Blank and Fedorenko, 2020*). The left IFG and insula are particularly associated with linguistic processes that require integration over longer periods of time, such as syntactic structure building and semantic integration of meaning (*Fedorenko et al., 2016*; *Matchin et al., 2017*; *Schell et al., 2017*), and are also recruited when speech comprehension requires effort, such as for degraded or noise-vocoded speech (*Davis and Johnsrude, 2003*; *Obleser and Kotz, 2010*; *Davis et al., 2011*; *Hervais-Adelman et al., 2012*). Accordingly, observing a phrase-level response to task-irrelevant speech in these regions is in line with their functional involvement in processing speech under adverse conditions.

With regard to the left posterior-parietal cluster, the interpretation for why a phrase-level response is observed there is less straightforward. Although some portions of the parietal cortex are involved in speech processing, these are typically more inferior than the cluster found here (*Hickok and Poeppel, 2007*; *Smirnov et al., 2014*). However, both the posterior-parietal cortex and inferior frontal gyrus play an important role in verbal working-memory (*Todd and Marois, 2004*; *Postle et al., 2006*; *Linden, 2007*; *McNab and Klingberg, 2008*; *Edin et al., 2009*; *Østby et al., 2011*; *Rottschy et al., 2012*; *Gazzaley and Nobre, 2012*; *Ma et al., 2012*; *Meyer et al., 2014*; *Meyer et al., 2015*; *Yue et al., 2019*; *Fedorenko and Blank, 2020*). Detecting the phrasal structure

of task-irrelevant speech, while focusing primarily on processing the to-be-attended narratives, likely requires substantial working-memory for integrating chunks of information over time. Indeed, attention and working-memory are tightly linked constructs (*McNab and Klingberg, 2008*; *Gazzaley and Nobre, 2012*; *Vandierendonck, 2014*), and as mentioned above, the ability to control and maintain attention is often associated with individual working-memory capacity (*Cowan et al., 2005*; *Beaman et al., 2007*; *Forster and Lavie, 2008*; *Naveh-Benjamin et al., 2014*; *Lambez et al., 2020*). Therefore, one possible interpretation for the presence of a phrase-level response to task-irrelevant speech in the left posterior-parietal cortex and inferior frontal regions, is their role in forming and maintaining a representation of task-irrelevant stimuli in working-memory, perhaps as a means for monitoring the environment for potentially important events.

## Why no word-level response?

Although in the current study we found significant neural response to task-irrelevant speech at the phrase-rate, we did not see peaks at the word- or at the sentence-rate. Regarding the sentence-level response, it is difficult to determine whether the lack of an observable peak indicates that the stimuli were not parsed into sentences, or if this null-result is due to the technical difficulty of obtaining reliable peaks at low-frequencies (0.5 Hz) given the 1/f noise-structure of neurophysiological recordings (*Pritchard, 1992*; *Miller et al., 2009*). Hence, this remains an open question for future studies. Regarding the lack of a word-level response at 2 Hz for Structured task-irrelevant stimuli, this was indeed surprising, since in previous studies using the same stimuli in a single-speaker context we observe a prominent peak at *both* the word- and the phrase-rate (*Makov et al., 2017*). Although we do not know for sure why the 2 Hz peak is not observed when this speech was presented as task-irrelevant concurrently with another narrative, we can offer some speculations for this null-result: One possibility is that the task-irrelevant speech was indeed parsed into words as well, but that the neural signature of 2 Hz parsing was not observable due to interference from the acoustic contributions at 2 Hz (see Supplementary Materials and *Luo and Ding, 2020*). However, another possibility is that the lack of a word-level response for task-irrelevant speech indicates that it does not undergo full lexical analysis. Counter to the linear intuition that syntactic structuring depends on identifying individual lexemes, there is substantial evidence that lexical and syntactic processes are separable and dissociable cognitive processes, that rely on partially different neural substrates (*Friederici and Kotz, 2003*; *Hagoort, 2003*; *Humphries et al., 2006*; *Nelson et al., 2017*; *Schell et al., 2017*; *Pylkkänen, 2019*; *Morgan et al., 2020*). Indeed, a recent frequency-tagging study showed that syntactic phrasal structure can be identified (generating a phrase-level peak in the neural spectrum) even in the complete absence of lexical information (*Getz et al., 2018*). Hence, it is possible that when speech is task-irrelevant and does not receive full attention, it is processed only partially, and that although phrasal boundaries are consistently detected, task-irrelevant speech does not undergo full lexical analysis. This matter regarding the depth of lexical processing of task-irrelevant speech, and its interaction with syntactic analysis, remains to be further explored in future research.

## Task-irrelevant influence on processing to-be-attended speech

Besides analyzing the frequency-tagged neural signatures associated with encoding the **task-irrelevant stimuli**, we also looked at how the neural encoding of **to-be-attended speech** was affected by the type of task-irrelevant speech it was paired with. In line with previous MEG studies, the speech-tracking response (estimated using TRFs) was localized to auditory temporal regions bilaterally and left inferior frontal regions (*Ding and Simon, 2012*; *Zion Golumbic et al., 2013a*; *Puvvada and Simon, 2017*). The speech tracking response in auditory regions was similar in both conditions; however, the response in left inferior-frontal cortex was modulated by the type of task-irrelevant speech presented and was enhanced when task-irrelevant speech was Structured vs. when it was Non-Structured. This pattern highlights the nature of the competition for resources triggered by concurrent stimuli. When the task-irrelevant stimulus was Non-Structured, even though it was comprised of individual phonetic-acoustic units, it did not contain meaningful linguistic information and therefore did not require syntactic and semantic resources. However, the Structured task-irrelevant speech poses more of a competition, since it constitutes fully intelligible speech. Indeed, it is well established that intelligible task-irrelevant speech causes more competition and therefore are more distracting than non-intelligible speech (*Rhebergen et al., 2005*; *Iyer et al., 2010*; *Best et al., 2012*; *Gallun and*

*Diedesch, 2013*; *Carey et al., 2014*; *Kilman et al., 2014*; *Swaminathan et al., 2015*; *Kidd et al., 2016*). A recent EEG study found that responses to both target and distractor speech are enhanced when the distractor was intelligible vs. unintelligible (*Olguin et al., 2018*), although this may depend on the specific type of stimulus used (*Rimmele et al., 2015*). However, in most studies it is difficult to ascertain the level(s) of processing where competing between the inputs occurs, and many effects can be explained by variation in the acoustic nature of maskers (*Ding and Simon, 2014*). The current study is unique in that all low-level features of Structured and Non-Structured speech stimuli were perfectly controlled, allowing us to demonstrate that interference goes beyond the phonetic-acoustic level and also occurs at higher linguistic levels. The findings that the speech tracking response of the to-be-attended narratives is enhanced when competing with a Structured task-irrelevant speech, specifically left inferior-frontal brain regions, where we also observed tracking of the phrase-structure of task-irrelevant speech, pinpoints the locus of this competition to these dedicated speech-processing regions, above and beyond any sensory-level competition (*Davis et al., 2011*; *Brouwer et al., 2012*; *Hervais-Adelman et al., 2012*). Specifically, they suggest that the enhanced speech tracking response in IFG reflects the investment of additional listening effort for comprehending the task-relevant speech (*Vandenberghe et al., 2002*; *Gagné et al., 2017*; *Peelle, 2018*).

Since the neural response to the to-be-attended speech was modulated by the type of competition it faced, then why was this not mirrored in the current behavioral results as well? In the current study participants achieved similar accuracy rates on the comprehension questions regardless of whether the natural-narratives were paired with Structured or Non-Structured stimuli in the task-irrelevant ear, and there was no significant correlation between the neural effects and performance. We attribute the lack of a behavioral effect primarily to the insensitivity of the behavioral measures used here, that consisted of asking four multiple-choice questions after each 45 s long narrative. Although numerous previous studies have been able to demonstrate behavioral 'intrusions' of the task-irrelevant stimuli on performance of an attended-task, these have been shown using more constrained experimental paradigms, that have the advantage of probing behavior at a finer scale, but are substantially less ecological (e.g. memory-recall for short lists of words or priming effects; *Tun et al., 2002*; *Dupoux et al., 2003*; *Rivenez et al., 2006*; *Rivenez et al., 2008*; *Carey et al., 2014*; *Aydelott et al., 2015*). In moving toward studying speech processing and attention under more ecological circumstances, using natural continuous speech, we face an experimental challenge of obtaining sufficiently sensitive behavior measures without disrupting listening with an ongoing task (e.g. target detection) or encroaching too much on working-memory. This is a challenge shared by many previous studies similar to ours, and is one of the main motivations for turning directly to the brain and studying neural activity during uninterrupted listening to continuous speech, rather than relying on sparse behavioral indications (*Ding et al., 2016*; *Makov et al., 2017*; *Brodbeck et al., 2018a*; *Broderick et al., 2018*; *Broderick et al., 2019*; *Donhauser and Baillet, 2020*).

## Conclusions

The current study contributes to ongoing efforts to understand how the brain deals with the abundance of auditory inputs in our environment. Our results indicate that even though top-down attention effectively enables listeners to focus on a particular task-relevant source of input (speech in this case), this prioritization can be affected by the nature of task-irrelevant sounds. Specifically, we find that when the latter constitutes meaningful speech, left fronto-temporal speech-processing regions are engaged in processing both stimuli, potentially leading to competition for resources and more effortful listening. Additional brain regions, such as the PPC, are also engaged in representing some aspects of the linguistic structure of task-irrelevant speech, which we interpret as maintaining a representation of what goes on in the 'rest of the environment', in case something important arises. Importantly, similar interactions between the structure of task-irrelevant sounds and responses to the to-be-attended sounds have been previously demonstrated for non-verbal stimuli as well (*Makov and Zion Golumbic, 2020*). Together, this highlights the fact that attentional selection is not an all-or-none processes, but rather is a dynamic process of balancing the resources allocated to competing input, which is highly affected by the specific perceptual, cognitive and environmental aspects of a given task.

## Acknowledgements

This work was funded by Binational Science Foundation (BSF) grant # 2015385 and ISF grant #2339/20. We would like to thank Dr. Nai Ding for helpful comments on a previous version of this paper.

## Additional information

### Funding

| Funder | Grant reference number | Author |
| --- | --- | --- |
| Israel Science Foundation | 2339/20 | Elana Zion Golumbic |
| United States - Israel Binational Science Foundation | 2015385 | Elana Zion Golumbic |

The funders had no role in study design, data collection and interpretation, or the decision to submit the work for publication.

### Author contributions

Paz Har-shai Yahav, Data curation, Formal analysis, Investigation, Visualization, Methodology, Writing - original draft, Writing - review and editing; Elana Zion Golumbic, Conceptualization, Resources, Data curation, Formal analysis, Supervision, Funding acquisition, Investigation, Methodology, Writing - original draft, Project administration, Writing - review and editing

### Author ORCIDs

Paz Har-shai Yahav (ID) https://orcid.org/0000-0002-3666-3338
Elana Zion Golumbic (ID) https://orcid.org/0000-0002-8831-3188

### Ethics

Human subjects:The study was approved by the IRB of Bar Ilan University on 1.1.2017, under the protocol titled "Linking brain activity to selective attention at a 'Cocktail Party' " (approval duration: 2 years). All participants provided written informed consent prior to the start of the experiment.

### Decision letter and Author response

Decision letter https://doi.org/10.7554/eLife.65096.sa1
Author response https://doi.org/10.7554/eLife.65096.sa2

## Additional files

### Supplementary files

• Transparent reporting form

### Data availability

The Full MEG data and examples of the stimuli are now available on the Open Science Framework repository (https://osf.io/e93qa).

The following dataset was generated:

| Author(s) | Year | Dataset title | Dataset URL | Database and Identifier |
| --- | --- | --- | --- | --- |
| Har-shai Yahav P, Zion Golumbic E | 2021 | Linguistic processing of task-irrelevant speech at a Cocktail Party | https://osf.io/e93qa | Open Science Framework, e93qa |

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

# Appendix 1

## Supplementary materials

The modulation spectrum of Structured speech stimuli used in this study featured a prominent peak at 4Hz, corresponding to the syllable-rate, and an additional smaller peak at 2Hz. Since the stimuli were built by concatenating 250-ms long syllables, while taking care not to introduce any additional acoustic events that would introduce other rhythmic regularities (such as systematic gaps between words; *Buiatti et al., 2009*), we hypothesized that it may be related to the order of the syllables within the Structured sequences. Specifically, since our Structured stimuli was comprised of bi-syllabic Hebrew words, there may be a systematic difference in the envelope-shape of syllables at the beginning vs. end of words. For example, in the materials used here, it was indeed more common to start a word with a CV syllable than to end with one (*Figure 1—figure supplement 1* and *2*). This, in turn, could lead to subtle yet systematic differences in the envelope-shape at even vs. odd positions in the stimulus, particularly after averaging across sentences/trials, resulting in a 2Hz peak in the modulation spectrum. A recent study by *Luo and Ding, 2020* nicely demonstrates that an acoustic-driven 2Hz peak can be induced simply by amplifying every second syllable in multi-syllable words.

To better understand the origin of the 2Hz peak in our Structured stimuli we ran several simulations, testing how the order of the syllables within the sequence affected the modulation spectrum. First, we created Position-Controlled Stimuli (*Figure 1—figure supplement 2b*), which were pseudo-sentences comprised of the same syllables as the Structured speech, but ordered in a manner that did not create linguistic meaning. Importantly, randomization was performed in a manner so that each syllable maintained the same position it had in the original sentences. For example, if the syllable /gu/ was the first in original Sentence #1 and the syllable /buk/ was the last, then in the Position-Controlled stimuli these two syllables will still be first or last, respectively, but will no longer be part of the same pseudo-sentence (see concrete examples in *Figure 1—figure supplement 2b*). In this manner, the average audio-envelope across sentences is still identical to the Structured materials, but the stimuli no longer carry linguistic meaning.

The same procedure was used for calculating the modulation spectrum as were applied to the original Structured stimuli (see Materials and methods for full details). Briefly, a total of 52 pseudo-sentences were constructed and randomly concatenated to form 50 sequences (56-seconds long). To stay faithful to the analysis procedure applied later to the MEG data, sequences were then divided into 8-second epochs, the envelope was extracted and FFT was applied to each epoch. The modulation spectrum is the result of averaging the spectrum across all epochs. We find that, indeed, the modulation spectrum of the original Structured materials and the Position-Controlled materials are basically identical, both containing similar peaks at 4Hz and 2Hz. This supports our hypothesis that the 2Hz peak stemmed from a natural asymmetry in the type of syllables that occur in start vs. end positions of bi-syllabic words (at least in Hebrew). We note that this type of Position-Controlled stimuli were used by us as Non-Structured stimuli in a previous study (*Makov et al., 2017*), and this is likely a more optimal choice as a control stimuli for future studies, as it allows to more confidently attribute differences at 2Hz in the neural response between Structured and Position-Controlled Non-Structured stimuli to linguistic, rather than acoustic, effects.

We next ran two additional simulations to determine what form of syllable randomization eliminates the 2Hz peak. We found that when creating pseudo-sentences where syllables are completely randomized and not constrained by position, the 2Hz peak is substantially reduced (*Figure 1—figure supplement 2c*). However, in this case we used a Fixed set of 52 pseudo-sentences to form sequences of stimuli. When we further relaxed this constraint, and allowed different randomization in different sequences (Non-Fixed Randomized stimuli; *Figure 1—figure supplement 2d*), the 2Hz peak was completely eliminated. The latter is akin to the Non-Structured stimuli used in the current experiment, and hence they were, in fact, not fully controlled at the acoustic level for 2Hz modulations.

That said, since we did not in fact see any peak in the neural data at 2Hz, and the effect we did find was at 1Hz (which is controlled across conditions), this caveat does not affect the validity of the results reported here. Future studies using this hierarchical frequency-tagging approach should take care to equate the modulation spectrum of experimental and control stimuli on this dimension as well (as was done by *Makov et al., 2017*).

