## [Decision Letter]

**Acceptance summary:**

Using a clever adaptation of a classic experimental paradigm, the authors assessed linguistic processing of task-irrelevant speech. By structuring the stimulus so that phonemes, words, phrases and sentence have nested but identifiable discrete rates. the authors were able to identify neural processing corresponding to each stimulus organizational level from magnetoencephalography responses. This study, which will interest those studying attention, language, or the organization of the auditory system, reveals linguistic processing of irrelevant speech at the phrasal level, which is an unexpected and intriguing result.

**Decision letter after peer review:**

Thank you for submitting your article "Linguistic processing of task-irrelevant speech at a Cocktail Party" for consideration by *eLife*. Your article has been reviewed by 3 peer reviewers, and the evaluation has been overseen by Barbara Shinn-Cunningham as the Senior and Reviewing Editor. The following individuals involved in review of your submission have agreed to reveal their identity: Phillip E Gander (Reviewer #1); Ross K Maddox (Reviewer #2).

Essential revisions:

1. It is surprising that your results do not replicate previous results showing peaks related to sentence- and word-level frequencies. Ideally, some other control experiment would be performed to directly address this issue. If such controls cannot be performed, the claims of the paper must be toned down.

2. The main effects is a change in a small peak that appears at 1 Hz, the frequency of the phrasal structure of the to-be-ignored speech. However, the reviewers raised a number of issues about the reliability of this peak measurement. Further analyses are needed to convince that the peak is real, especially given that the measured spectra are noisy-such as a quantification of effect sizes, quantification of the modulation spectrum of the natural, to-be-attended speech (and a consideration of how this interacts with the manipulated spectrum of the to-be-ignored speech), or related analyses.

3. Related to point 2, please clarify exactly how the ITPC normalized and how this relates to the noise floor in your measures (see Reviewer 3's remarks).

4. To strengthen your claims, please explore whether individual subject data are correlated when you compare response accuracy in the structured condition with the strength of the phrasal-level ITPC. If there is no significant relationship, please offer some interpretation as to why there is not.

5. Lapses of attention that are not especially prolonged could contribute to or even explain some of your results. While you mention this and are careful with your language, your Discussions should consider this issue more fully and fairly.

6. Please discuss how "natural" the manipulated speech stimuli were, and whether this differed across conditions (or is different in important ways compared to previous studies). Do you believe any such differences explain what you observed?

In resubmitting your paper, please also consider the many suggestions and questions that the reviewers raise in their full reviews, below.

*Reviewer #1:*

The present study sought to better characterize how listeners deal with competing speech streams from multiple talkers, that is, whether unattended speech in a multi talker environment competes for exclusively lower-level acoustic/phonetic resources or whether it competes for higher-level linguistic processing resources as well. The authors recorded MEG data and used hierarchical frequency tagging in an unattended speech stream presented to one ear while listeners were instructed to attend to stories presented in the other ear. The study found that when the irrelevant speech contained structured (linguistic) content, an increase in power at the phrasal level (1 Hz) was observed, but not at the word level (2 Hz) or the sentence level (0.5 Hz). This suggests that some syntactic information in the unattended speech stream is represented in cortical activity, and that there may be a disconnect between lexical (word level) processing and syntactic processing. Source analyses of the difference between conditions indicated activity in left inferior frontal and left posterior parietal cortices. Analysis of the source activity underlying the linear transformation of the stimulus and response revealed activation in left inferior frontal (and nearby) cortex. Implications for the underlying mechanisms (whether attentional shift or parallel processing) are discussed. The results have important implications for the debate on the type and amount of representation that occurs to unattended speech streams.

The authors utilize clever tools which arguably provided a unique means to address the main research question, i.e., they used hierarchical frequency tagging for the distractor speech, which allowed them to assess linguistic representations at different levels (syllable-, word-, phrase-, and sentence-level). This technique enabled the authors to make claims about what level of language hierarchy the stimuli are being processed, depending on the observed frequency modulation in neural activity. These stimuli were presented during MEG recording, which let the authors assess changes in neurophysiological processing in near real time – essential for research on spoken language. Source analyses of these data provided information on the potential neural mechanisms involved in this processing. The authors also assessed a temporal response function (TRF) based on the speech envelope to determine the brain regions involved at these different levels for linguistic analysis of the distractor speech.

1. Speech manipulation:

In general, it is unclear what predictions to make regarding the frequency tagging of the unattended distractor speech. On the one hand, the imposed artificial rhythmicity (necessary for the frequency tagging approach) may make it easier for listeners to ignore the speech stream, and thus seeing an effect at higher-level frequency tags may be of greater note, although not entirely plausible. On the other hand, having the syllables presented at a consistent rate may make it easier for listeners to parse words and phrasal units because they know precisely when in time a word/phrase/sentence boundary is going to occur, allowing listeners to check on the irrelevant speech stream at predictable times. For both the frequency tagging and TRF electrophysiological results, the task-irrelevant structured speech enhancement could be interpreted as an infiltration of this information in the neural signal (as the authors suggest), but because the behavioral results are not different this latter interpretation is not easily supported. This pattern of results is difficult to interpret.

2. Behavioral Results:

Importantly, no behavioral difference in accuracy was observed between the two irrelevant speech conditions (structured vs. non-structured), which makes it difficult to interpret what impact the structured irrelevant speech had on attentive listening. If the structured speech truly "infiltrates" or "competes" for linguistic processing resources, the reader would assume a decrease in task accuracy in the structured condition. This behavioral pattern has been observed in other studies. This calls into questions the face validity of the stimuli and task being used.

3. Attention:

• In this study activation of posterior parietal cortex was found, that could be indicative of a strong attentional manipulation, and that the task was in fact quite attentionally demanding in order for subjects to perform. This may align with the lack of behavioral difference between structured and non-structured irrelevant stimuli. Perhaps subjects attempted to divide their attention which may have been possible between speech that was natural and speech that was rather artificial. The current results may align with a recent proposal that inferior frontal activity may be distinguished by language selective and domain general patterns.

4. Lack of word level response:

• A major concern is that the results do not seem to replicate from an earlier study with the same structured stimuli, i.e., the effects were seen for sentence and word level frequency tagging. As the authors discuss, it seems difficult to understand how a phrasal level of effect could be obtained without word-level processing, and so a response at the word level is expected.

5. Familiarization phase:

The study included a phase of familiarization with the stimuli, to get participants to understand the artificial speech. However it would seem that it is much easier for listeners to report back on structured rather than unstructured stimuli. This is relevant to understanding any potential differences between the two conditions. It is unclear if any quantification was made of performance/understanding at this phase. If there is no difference in the familiarization phase, this might explain why there was no difference in behavior during the actual task between the two conditions. Or, if there is a difference at the familiarization phase (i.e. structured sequences are more easily repeated back than non-structured sequences), this might help explain the neural data result at 1 Hz, given that some higher level of processing must have occurred for the structured speech (such as "chunking" into words/phrasal units).

6. Speech manipulation:

• To assist interpretation of the pattern of results it would be helpful to know something about the listening experience to the stimuli. Was there any check on abnormality of the attended speech that was changed in gender, i.e., how natural did this sound? Why was the gender manipulation imposed, instead of initially recording the stimuli in a male voice?

• Similarly, how natural-sounding was the task-irrelevant structured speech?

7. Behavioral Results:

• It is difficult to reconcile why there are no significant differences in response accuracy between non-structured and structured conditions. How do the authors interpret this discrepancy alongside the observed neural differences across these two conditions? Is it possible this is reflective of the stimulus, i.e., that the structured speech is so artificial that it is not intrusive on the attended signal (thus no difference in behavior)? It seems less plausible that the non-structured speech was equally as intrusive as the structured speech, given the cited literature. These issues relate to the speech manipulation.

• Did the authors check on task behavior before running the experiment?

• How does this relate to the irrelevant stimuli were used in the experiment in Makov et al. 2017. Is it possible this is important, can the authors comment?

• It would be helpful to understand why no behavioral result occurred, and whether this was related to the speech manipulation specifically by obtaining behavioral results with other versions of the task.

• There is a rather large range in response accuracy across subjects. One analysis that might strengthen the argument that the increase in 1 Hz ITPC truly reflects phrasal level information would be to look for a correlation between response accuracy in the structured condition with 1 Hz ITPC. One might predict that listeners which show lower behavioral accuracy would show greater 1 Hz ITPC (i.e. greater interference from linguistic content of the unattended structured speech).

8. Attention:

• An attentional load account is interesting, but load is not directly manipulated in this experiment. And it is difficult to reconcile the claim that the Ding experiment was less natural than the current one, as both were transformations of time.

• Federenko and Blank, 2020 propose an account of inferior frontal mechanisms that may relate to the present pattern of results regarding, in particular that there may be a stronger manipulation of attention (domain general) than linguistic processes.

9. Lack of word level response:

• Again it might be relevant with respect to potential stimulus differences to previous versions of the stimuli. If this is the case then it might have important implications for only a 1Hz effect being found.

• If SNR is an issue it calls into question the current results in addition to the possibility that it was an underestimation.

10. Familiarization phase:

• If they exist, what are the behavioral data / accuracy between conditions (structured v. non-structured) from the familiarization phase? Or did subjects comment on any differences?

11. ROIs:

• A representation or list of the ROIs might be helpful. It is unclear from Figure 1 if the whole cortical mantle is included in the 22 ROIs. In addition these ROI boundaries look considerably larger than the 180 in each hemisphere from Glasser et al. 2016. Please clarify.

*Reviewer #2:*

This paper by Har-shai Yahav and Zion Golumbic investigates the coding of higher level linguistic information in task-irrelevant speech. The experiment uses a clever design, where the task-irrelevant speech is structured hierarchically so that the syllable, word, and sentence levels can be ascertained separately in the frequency domain. This is then contrasted with a scrambled condition. The to-be-attended speech is naturally uttered and the response is analyzed using the temporal response function. The authors report that the task-irrelevant speech is processed at the sentence level in the left fronto-temporal area and posterior parietal cortex, in a manner very different from the acoustical encoding of syllables. They also find that the to-be-attended speech responses are smaller when the distractor speech is not scrambled, and that this difference shows up in exactly the same fronto-temporal area – a very cool result.

This is a great paper. It is exceptionally well written from start to finish. The experimental design is clever, and the results were analyzed with great care and are clearly described.

The only issue I had with the results is that the possibility (or likelihood, in my estimation) that the subjects are occasionally letting their attention drift to the task-irrelevant speech rather than processing in parallel can't be rejected. To be fair, the authors include a nice discussion of this very issue and are careful with the language around task-relevance and attended/unattended stimuli. It is indeed tough to pull apart. The second paragraph on page 18 states "if attention shifts occur irregularly, the emergence of a phase-rate peak in the neural response would indicate that bits of 'glimpsed' information are integrated over a prolonged period of time." I agree with the math behind this, but I think it would only take occasional lapses lasting 2 or 3 seconds to get the observed results, and I don't consider that "prolonged." It is, however, much longer than a word, so nicely rejects the idea of single-word intrusions.

*Reviewer #3:*

The use of frequency tagging to analyze continuous processing at phonemic, word, phrasal and sentence-levels offers a unique insight into neural locking at higher-levels. While the approach is novel, there are major concerns regarding the technical details and interpretation of results to support phrase-level responses to structured speech distractors.

– Is the peak at 1Hz real and can it be attributed solely to the structured distractor?

* The study did not comment on the spectral profile of the "attended" speech, and how much low modulation energy is actually attributed to the prosodic structure of attended sentences? To what extent does the interplay of the attended utterance and distractor shapes the modulation dynamics of the stimulus (even dichotically)?

* How is the ITPC normalized? Figure 2 speaks of a normalization but it is not clear how? The peak at 1Hz appears extremely weak and no more significant (visually) than other peaks – say around 3Hz and also 2.5Hz in the case of non-structured speech? Can the authors report on the regions in modulation space that showed any significant deviations? What about effect size of the 1Hz peak relative to these other regions?

* It is hard to understand where the noise floor in this analysis – this floor will rotate with the permutation test analysis performed in the analysis of the ITPC and may not be fully accounted for. This issue depends on what the chosen normalization procedure is. The same interpretation put forth by the author regarding a lack of a 0.5Hz peak due to noise still raises the question of interpreting the observed 1Hz peak?

– Control of attention during task performance

* The author present a very elegant analysis of possible alterative accounts of the results, but they acknowledge that possible attention switches, even if irregular, could result in accumulated information that could emerge as a small neurally-locked response at the phrase-level? As indicated by the authors, the entire experimental design to fully control for such switches is a real feat. That being said, additional analyses could shed some light on variations of attentional state and their effect on observed results. For instance, analysis of behavioral data across different trials (wouldn't be conclusive, but could be informative)

* This issue is further compounded by the fact that a rather similar study (Ding et al.) did not report any phrasal-level processing, though there are design differences. The authors suggest differences in attentional load as a possible explanation and provide a very appealing account or reinterpretation of the literature based on a continuous model of processing based on task demands. While theoretically interesting, it is not clear whether any of the current data supports such account. Again, maybe a correlation between neural responses and behavioral performance in specific trials could shed some light or strengthen this claim.

– What is the statistic shown for the behavioral results? Is this for the multiple choice question? Then what is the t-test on?

– Beyond inter-trial phase coherence, can the authors comment on actual power-locked responses at the same corresponding rates?

In line with concerns regarding interpretation of experimental findings, some control experiment appears to be critical to establish a causal link between the observed neural processing the 1Hz rhythm and the phrasal processing of the distractor. What do the author expect from a similarly structured distractor but unintelligible, in an unfamiliar language or even reversed?

---

## [Author Response]

Essential revisions:1. It is surprising that your results do not replicate previous results showing peaks related to sentence- and word-level frequencies. Ideally, some other control experiment would be performed to directly address this issue. If such controls cannot be performed, the claims of the paper must be toned down.

Indeed, the lack of a response at the word-level was not expected. All the reviewers point this out and we too try to grapple with this issue. We too expected, based on previous findings, that if linguistic-responses were observed for task-irrelevant speech, this would be observed at *both* the word- and phrase- levels. Indeed, this is the case when participants listen to these stimuli from single-speaker, without competition; e.g., Ding et al. 2016, Makov et al. 2017. As detailed below in our response to reviewers #1 and #3 and revised methods and Results section, we conducted additional analyses of the data to confirm the validity of the 1Hz peak (despite the lack of a 2Hz peak), and this result still stands. Since these are the data, which we now share in full on OSF (https://osf.io/e93qa), we offer some thoughts as to why the word-level peak is not observed when frequency-tagged speech is the ‘task-irrelevant’ stimulus in a two-speaker paradigm (as opposed to a single, attended speaker). For a more elaborate response see section “why no word-level response” in our revised paper (Discussion, p. 23), and our detailed response to reviewers #1 and #3 below.

2. The main effects is a change in a small peak that appears at 1 Hz, the frequency of the phrasal structure of the to-be-ignored speech. However, the reviewers raised a number of issues about the reliability of this peak measurement. Further analyses are needed to convince that the peak is real, especially given that the measured spectra are noisy-such as a quantification of effect sizes, quantification of the modulation spectrum of the natural, to-be-attended speech (and a consideration of how this interacts with the manipulated spectrum of the to-be-ignored speech), or related analyses.

We understand the reviewers’ concerns about the validity of the 1Hz peak, and we have several responses to this comment:

1. We have added a new analysis to independently assess the significance of the 1Hz peak. In this analysis, we compared ITPC at each frequency to the average ITPC in the surrounding frequencies (2 bins from each side) using a t-test. This approach addresses the concern of frequency-specific noise-floor, for example due to the inherent 1/f noise structure (see similar implementations e.g., in Mouraux et al., 2011; Retter and Rossion, 2016, Nozaradan et al. 2017, 2018). Results of this analysis confirmed that the only significant ITPC peaks were at 4Hz (in both conditions) and at 1Hz (in the Structured condition only), further validating the robustness of the 1Hz phrase-level response. Note that the other peaks that stand-out visually (at 2.5Hz and 3Hz) were not statistically significant in this analysis. The new analysis is now described in the methods (pp. 11) and Results sections (pp. 14).

2. Although the magnitude of the 1Hz peak is smaller than the 4Hz, this is to be expected since the 4Hz peak is a direct consequence of the acoustic input whereas the 1Hz peak proposedly reflects linguistic representation and/or chunking. Similar ratios between the acoustic and linguistic peaks have been observed in previous studies using this frequency-tagging approach (e.g., Ding et al. 2016, Makov et al. 2017). Therefore, the smaller relative magnitude of the 1Hz peak does not invalidate it. We have now added quantification of all effect sizes to the Results section.

3. The reviewers suggests that, perhaps, the acoustics of the task-relevant stream could have contributed to the observed 1Hz ITPC peak, and that the peak might not solely be attributed to the Structured stimuli. However, this is not the case for two main reasons:

a. In our response to reviewer #3 we show the modulation spectrum of the narratives used as task-relevant speech. As can be seen, the spectrum does not contain a distinct peak at 1Hz.

b. The narratives used as task-relevant speech were fully randomized between the Structured and Non-Structured conditions and between participants. Therefore, if these stimuli had contributed to the 1Hz peak, this should have been observed in both conditions. We have now clarified this point in the methods (p.6).

3. Related to point 2, please clarify exactly how the ITPC normalized and how this relates to the noise floor in your measures (see Reviewer 3's remarks).

The normalization used for the ITPC was a z-score of the resultant phase-locking factor, as implemented in the circ_rtest function in the Matlab circular statistics toolbox (Berens, 2009). We have now clarified this in the methods section (p. 11) and the caption of Figure 3 (p. 14).

4. To strengthen your claims, please explore whether individual subject data are correlated when you compare response accuracy in the structured condition with the strength of the phrasal-level ITPC. If there is no significant relationship, please offer some interpretation as to why there is not.

To test whether the 1Hz ITPC peak was stronger in participants who performed poorly we have added a new analysis. We split the participants into two groups according to the median of the 1Hz ITPC value in the Structured condition and tested whether there was a significant difference in the behavioral scores of the two groups. However, this median-split analysis did not reveal any significant effects that would indicate a ‘trade-off’ between linguistic representation of task-irrelevant speech and performance on the attended task. That said, as we elaborate in our response to reviewer #1/critique #2 below and in the revised methods and discussion, the task used here (answering 4 multiple-choice questions after each 45-second long narrative) was not sufficiently sensitive for adequately capturing behavioral consequences of task-irrelevant stimuli. Therefore, in our opinion, this null-effect should *not* necessarily be taken as indication that both types of task-irrelevant stimuli were similarly distracting. We have added this new analysis to the paper (p. 13-14).

5. Lapses of attention that are not especially prolonged could contribute to or even explain some of your results. While you mention this and are careful with your language, your Discussions should consider this issue more fully and fairly.

Indeed, as the reviewers points out, one possible interpretation for the current results is that the strict rhythmic/hierarchical structure of task-irrelevant speech-stimuli enabled participants to employ a ‘rhythmic-glimpsing listening strategy’. In this strategy the task irrelevant stream could be sampled (‘glimpsed’) at the most informative points in time, allowing participants to glean sufficient linguistic information to process the task-irrelevant speech. We agree (and discuss in the paper; pp. 20) that the phrase-rate peak observed here in the Structured condition could, potentially, reflect this type of ‘glimpsing’ strategy (even though we cannot validate or disprove this possibility in the current study, since we do not have direct insight into participants’ listening strategy).

However, even if this is the case, employing a ‘rhythmic glimpsing strategy’ is in-and-of-itself an indication that the linguistic structure of the task-irrelevant stream was detected and is parsed correctly. This is because, given the hierarchical design of our Structured stimuli, for participants to "know precisely when in time a word/phrase/sentence boundary is going to occur" they must integrate over several syllables to identify the underlying linguistic structure. This is not trivial since the linguistic structure does not follow directly from the acoustics. Also, since we gradually ramped up the volume of the task-irrelevant speech at the start of each trial, determining where each sentence/phrase starts can only be achieved based on linguistic processing. Therefore, finding a 1Hz peak for the Structured but not for the Non-Structured task-irrelevant speech, which share the same acoustic-rhythm and were constructed from the same syllable units, serves as an indication that the underlying linguistic structure of task irrelevant speech was indeed detected. We have elaborated on this point in the revised discussion (pp. 20).

6. Please discuss how "natural" the manipulated speech stimuli were, and whether this differed across conditions (or is different in important ways compared to previous studies). Do you believe any such differences explain what you observed?

This is an important point, since our ultimate goal is to understand attention to speech under natural, real-life, conditions. We agree that the current study is not fully ‘natural’ for several reasons (e.g. the structured nature of the stimuli; their arbitrary content; the dichotic presentation etc.). The revised discussion now addressed this point more extensively, focusing on the generalization of our findings to more "natural” circumstances, comparison to previous studies, and a call for future studies to continue in this direction and systematically explore these questions under increasingly natural conditions (p. 21-22 and p.25-26; see also specific response to the reviewers below):

“As cognitive neuroscience research increasingly moves towards studying speech processing and attention in real-life circumstances, a critical challenge will be to systematically map out the perceptual and cognitive factors that contribute to, or hinder, the ability to glean meaningful information from stimuli that are outside the primary focus of attention.”In resubmitting your paper, please also consider the many suggestions and questions that the reviewers raise in their full reviews, below.Reviewer #1:[…] 1. Speech manipulation:In general, it is unclear what predictions to make regarding the frequency tagging of the unattended distractor speech. On the one hand, the imposed artificial rhythmicity (necessary for the frequency tagging approach) may make it easier for listeners to ignore the speech stream, and thus seeing an effect at higher-level frequency tags may be of greater note, although not entirely plausible. On the other hand, having the syllables presented at a consistent rate may make it easier for listeners to parse words and phrasal units because they know precisely when in time a word/phrase/sentence boundary is going to occur, allowing listeners to check on the irrelevant speech stream at predictable times. For both the frequency tagging and TRF electrophysiological results, the task-irrelevant structured speech enhancement could be interpreted as an infiltration of this information in the neural signal (as the authors suggest), but because the behavioral results are not different this latter interpretation is not easily supported. This pattern of results is difficult to interpret.

Although the current results provide an indication for linguistic-parsing of task-irrelevant speech, the reviewer raises an important point regarding the generalizability of our results to natural speech. Specifically, they ask whether the strict rhythmicity of the frequency tagged stimuli used here, might have made it easier or harder to "ignore", relative to natural speech that lacks this precise temporal structure.

We have several responses to this comment:

1. Indeed, as the reviewer points out, one possible interpretation for the current results is that the strict rhythmic/hierarchical structure of task-irrelevant speech-stimuli enabled participants to employ a ‘rhythmic-glimpsing listening strategy’. In this strategy the task irrelevant stream could be sampled (‘glimpsed’) at the most informative points in time, allowing participants to glean sufficient linguistic information to process the task irrelevant speech. We agree (and discuss in the paper; pp. 20) that the phrase-rate peak observed here in the Structured condition could, potentially, reflect this type of ‘glimpsing’ strategy (even though we cannot validate or disprove this possibility in the current study, since we do not have direct insight into participants’ listening strategy).

2. However, even if this is the case, employing a ‘rhythmic glimpsing strategy’ is in-and-of itself an indication that the linguistic structure of the task-irrelevant stream was detected and is parsed correctly. This is because, given the hierarchical design of our Structured stimuli, for participants to "know precisely when in time a word/phrase/sentence boundary is going to occur" they must integrate over several syllables to identify the underlying linguistic structure. This is not trivial, since the linguistic structure does not follow directly from the acoustics. Also, since we gradually ramped up the volume of the task-irrelevant speech at the start of each trial, determining where each sentence/phrase starts can only be achieved based on linguistic processing. Therefore, finding a 1Hz peak for the Structured but not for the Non-Structured task-irrelevant speech, which share the same acoustic-rhythm and were constructed from the same syllable units, serves as an indication that the underlying linguistic structure of task irrelevant speech was indeed detected.

We have elaborated on this point in the revised discussion, which now reads (pp. 20):

“An important question to ask is whether the phrase-level response observed for task irrelevant speech can be explained by attention shifts? […] However, we hope that future studies will gain empirical access into the dynamic of listeners’ internal attentional state and help shed light on this pivotal issue.”

3. Another important point raised by the reviewer is that the artificial-rhythmicity of the current stimuli might actually make them easier to ignore than natural speech that is less rhythmic? Indeed, as the reviewer points out, previous studies using simple tones suggest that when task-irrelevant tones are isochronous, this can make them easier to ignore relative to non-isochronous tones (e.g., Makov et al. 2020). The current study was not designed to address this particular aspect; since we do not compare whether isochronous speech is easier to ignore than non-isochronous speech. However, given the monotonous and artificial nature of the current speech-stimuli we speculate that they are probably easier to ‘tune out’, as compared to natural speech that contain potential attention grabbing events as well as prosodic cues. A recent study by Aubanel and Schwarz (2020) also suggests that isochrony plays a reduced role in speech perception relative to simple tones. Importantly, though, regarding the reviewer’s concern: the isochronous 4Hz nature of the speech-stimuli is not sufficient for explaining the main 1Hz effect reported here, since both the Structured and Non-Structured stimuli were similarly isochronous. Therefore, although we agree that this speech is “unnatural”, it is nonetheless processed and parsed for linguistic content.

4. Taking a broader perspective to the reviewer’s comment regarding the generalization of our findings to more "natural” circumstances:

One of the main take-home-messages of this paper is that we should not think about whether or not task-irrelevant speech is processed linguistically as a binary yes/no question. Rather, that the ability to process task-irrelevant speech likely depends on both the acoustic properties of the stimuli as well as the cognitive demands of the task. Within this framework, our current findings nicely demonstrate one of the circumstances in which task-irrelevant speech has the capacity to process more than one speech stream (be it through glimpsing or parallel processing, as discussed above). This demonstration has important theoretical implications, indicating that we should not assume an inherent “bottleneck” for processing two competing speech-streams, as suggested by some models.

Clearly, additional research is required to fully map-out the factors contributing to this effect (rhythmicity being one of them, perhaps). However, it is highly unlikely that the system is ONLY capable of doing this when stimuli are strictly rhythmic. We look forward to conducting systematic follow-up studies into the specific role of rhythm in attention to speech, as well as other acoustic and cognitive factors. We now elaborate on this point in the discussion (p. 21):

“More broadly, the comparison between these two studies invites re-framing of the question regarding the type / level of linguistic processing applied to task-irrelevant speech, and propels us to think about this issue not as a yes-or-no dichotomy, but perhaps as a more flexible process that depends on the specific context (Brodbeck et al. 2020b). [….] As cognitive neuroscience research increasingly moves towards studying speech processing and attention in real-life circumstances, a critical challenge will be to systematically map out the perceptual and cognitive factors that contribute to, or hinder, the ability to glean meaningful information from stimuli that are outside the primary focus of attention.”2. Behavioral Results:Importantly, no behavioral difference in accuracy was observed between the two irrelevant speech conditions (structured vs. non-structured), which makes it difficult to interpret what impact the structured irrelevant speech had on attentive listening. If the structured speech truly "infiltrates" or "competes" for linguistic processing resources, the reader would assume a decrease in task accuracy in the structured condition. This behavioral pattern has been observed in other studies. This calls into questions the face validity of the stimuli and task being used.

In the current study, attentive-listening behavior was probed by asking participants four multiple-choice questions after each narrative (3-possible answers per question; chance level = 0.33). Indeed, we did not find any differences in accuracy on these questions when the natural-narrative was paired with Structured vs. Non-Structured task-irrelevant speech. Moreover, **a new analysis aimed at testing whether behavioral performance was modulated by the strength of the 1Hz ITPC response did not reveal any significant effects (see behavioral results, pp. 14). However, we were not surprised by the lack of a behavioral effect, nor do we believe that this null-effect diminishes the significance of the neural effect observed here.** This is because the task was not designed with the intent of demonstrating behavioral disturbance-effects by task-irrelevant speech, and indeed does not have sufficient sensitivity to do so for several reasons:

a. Asking only four questions on a ~45-second long narrative is an extremely poor measure for probing a participant’s understanding of the entire narrative.

b. Answering these questions correctly or incorrectly is not necessarily a direct indication of how much attention was devoted to the narrative, or an indication of lapses of attention. Rather, performance is likely influenced by other cognitive processes. For example, questions requiring recollection of specific details (e.g., “what color was her hat?”) rely not only on attention but also on working-memory. Conversely, questions addressing the ‘gist’ of the narrative, (e.g., “why was she sad?”) can be answered correctly based on logical deduction even if there were occasional attention-lapses. In other words, good performance on this task does not necessarily indicate “perfect” attention to the narrative, just as making mistakes on this task does not necessarily reflect “lapses” of attention.

c. Supporting our claim of the coarseness of this task: Prior to this experiment, we conducted a behavioral pilot study, aimed at obtaining a baseline-measure for performance when participants listen to these narratives in a single-speaker context (without additional competing speech). In that study, the average accuracy rate was 83% (n=10), indicating that even without the presence of competing speech, performance on this task is not perfect.

So why did we choose this task?

As we discuss in the paper, and the reviewer correctly points out, there have been numerous behavioral studies demonstrating behavioral ‘intrusions’ of the task-irrelevant stimuli on performance of an attended-task. However, these effects require substantially more sensitive behavioral tasks, where behavior is probed at a finer scale. Some prominent examples are short-term memory-tasks for lists of words (Tun et al. 2002), semantic priming (Dupoux et al. 2003, Aydelott et al. 2015) or target-detection tasks (Rivenez et al. 2006, 2008, Carey et al. 2014). However, these types of tasks are quite artificial and are not suitable for studying attention to continuous natural speech particularly if we do not want to disrupt listening with an ongoing (secondary) task and/or artificial manipulation of the speech. Therefore, many studies similar to ours employ ‘gross’ comprehension metrics, that are by-definition inadequate for capturing the complexity of attentive-listening behavior.

Our main motivation for choosing this task was a) to motivate and guide participants to direct attention towards the to-be-attended narrative and b) verify that indeed they listened to it. This is critical in order to assert that participants indeed attended to the correct stream.

However, given the admitted insensitivity of this measure, we do not believe that the null effects on behavior can be interpreted in any meaningful way. In fact, the lack of good behavioral metrics for studying attention to continuous speech, and the tension of determining the ‘ground truth’ of attention is precisely the motivation for this study. We believe that ongoing neural metrics provide a better indication of how continuous speech is processed than sporadic assessment of comprehension/memory.

We now elaborate on this point in the paper in the methods section (p. 9):

“This task was chosen as a way to motivate and guide participants to direct attention towards the to-be-attended narrative and provide verification that indeed they listened to it. […] At the same time, this task is instrumental in guiding participants' selective attention toward the designated speaker, allowing us to analyze their neural activity during uninterrupted listening to continuous speech, which was the primarily goal of the current study.”

And in the discussion (p. 25):

“Since the neural response to the to-be-attended speech was modulated by the type of competition it faced, then why was this not mirrored in the current behavioral results as well? […] This is a challenge shared by many previous studies similar to ours, and is one of the main motivations for turning directly to the brain and studying neural activity during uninterrupted listening to continuous speech, rather than relying on sparse behavioral indications (Ding et al. 2016; Makov et al. 2017; Broderick et al. 2018, 2019; Donhauser and Baillet 2019; Brodbeck et al. 2020).”3. Attention:• In this study activation of posterior parietal cortex was found, that could be indicative of a strong attentional manipulation, and that the task was in fact quite attentionally demanding in order for subjects to perform. This may align with the lack of behavioral difference between structured and non-structured irrelevant stimuli. Perhaps subjects attempted to divide their attention which may have been possible between speech that was natural and speech that was rather artificial. The current results may align with a recent proposal that inferior frontal activity may be distinguished by language selective and domain general patterns.

We agree with the reviewer processing aspects of the task-irrelevant speech, in addition to following the to-be-attended narrative may impose a higher demand on working-memory (a form of ‘divided’ attention), and this might be reflected in the activation of PPC and inferior frontal regions in this condition. We have now expanded on this in our discussion, which reads (Discussion p. 22-23):

“Both the posterior-parietal cortex and inferior frontal gyrus play an important role in verbal working-memory (Todd and Marois 2004; Postle et al. 2006; Linden 2007; McNab and Klingberg 2008; Edin et al. 2009; Østby et al. 2011; Gazzaley and Nobre 2012; Ma et al. 2012; Rottschy et al. 2012; Meyer et al. 2014, 2015; Yue et al. 2019, Fedorenko and Blank 2020). […] Therefore, one possible interpretation for the presence of a phrase-level response to task-irrelevant speech in the left posterior-parietal cortex and inferior frontal regions, is their role in forming and maintaining a representation of task-irrelevant stimuli in working-memory, perhaps as a means for monitoring the environment for potentially important events.”4. Lack of word level response:• A major concern is that the results do not seem to replicate from an earlier study with the same structured stimuli, i.e., the effects were seen for sentence and word level frequency tagging. As the authors discuss, it seems difficult to understand how a phrasal level of effect could be obtained without word-level processing, and so a response at the word level is expected.

Indeed, the lack of a response at the word-level was not expected. We too expected that if linguistic-responses were observed for task-irrelevant speech, this would be observed at both the word- and phrase- level (as was observed in previous single-speaker studies using these stimuli; e.g., Makov et al. 2017). All the reviewers point this out and we too try to grapple with this issue in our discussion. As detailed above, we conducted additional analyses of the data to confirm that validity of the 1Hz peak (and the lack of a 2Hz peak), and this result still stands. **Since these are the data**, which we now share in full on OSF (https://osf.io/e93qa), we offer our thoughts as to why there is only a phrase-level peak but not a word-level peak when frequency-tagged speech is the ‘task-irrelevant’ stimulus in a two-speaker paradigm. See section “why no word-level response” in our revised paper (Discussion, section p. 23):

**“Why no word-level response?**Although in the current study we found significant neural response to task-irrelevant speech at the phrasal-rate, we did not see peaks at the word- or at the sentence-rate. […] This matter regarding the depth of lexical processing of task-irrelevant speech, and its interaction with syntactic analysis, remains to be further explored in future research.”

We understand that the lack of the expected 2Hz response can raise doubts regarding the validity of the 1Hz response as well. To address this concern and further verify that the peak observed at 1Hz is “real”, we have now added a new analysis to independently assess the significance of the 1Hz peak relative to the frequency-specific noise-floor. To this end, we compared ITPC at each frequency to the average ITPC in the surrounding frequencies (2 bins from each side) using a t-test. This approach accounts for potential frequency-specific variations in SNR, for example due to the inherent 1/f noise-structure (Mouraux et al., 2011; Retter and Rossion, 2016; Nozaradan et al., 2017, 2018). This analysis confirmed that the only significant ITPC peaks were at 4Hz (in both conditions) and at 1Hz (in the Structured condition only), further validating the robustness of the 1Hz phrase-level response. The new analysis is now described in the methods (pp. 11) and Results sections (pp. 14) which reads:

“Scalp-level spectra of the Inter-trial phase coherence (ITPC) showed a significant peak at the syllabic-rate (4Hz) in response to both Structured and Non-Structured hierarchical frequency tagged speech, with a 4-pole scalp-distribution common to MEG recorded auditory responses (Figure 3a) (p< 10^-9; large effect size, Cohen's d > 1.5 in both). […] Comparison of the 1Hz ITPC between these conditions also confirmed a significant difference between them (p=0.045; moderate effect size, Cohen's d = 0.57).”5. Familiarization phase:The study included a phase of familiarization with the stimuli, to get participants to understand the artificial speech. However it would seem that it is much easier for listeners to report back on structured rather than unstructured stimuli. This is relevant to understanding any potential differences between the two conditions. It is unclear if any quantification was made of performance/understanding at this phase. If there is no difference in the familiarization phase, this might explain why there was no difference in behavior during the actual task between the two conditions. Or, if there is a difference at the familiarization phase (i.e. structured sequences are more easily repeated back than non-structured sequences), this might help explain the neural data result at 1 Hz, given that some higher level of processing must have occurred for the structured speech (such as "chunking" into words/phrasal units).

We would like to clarify the need of the familiarization stage. The speech materials used in the current study are not immediately recognizable as Hebrew speech. Therefore, we conducted a familiarization stage prior to the main experiment. Without the familiarization stage, we could not be sure whether participants were aware of higher-level chunking of the Structured stimuli. Moreover, we wanted to avoid any perceptual learning effects during the main experiment. We now elaborate more extensively on the familiarization procedure in the methods section (pp. 9).

Indeed, as the reviewer anticipated, repeating the Structured stimuli during the familiarization stage was easier than repeating the Non-Structured stimuli since the latter did not map only known lexical units. However, we don’t fully understand the reviewers concern regarding the familiarization task and whether it ‘explains’ the neural results. As we see it, the 1Hz neural response reflects the identification of phrasal boundaries, which was only possibly for the Structured stimuli. The fact that participants were (implicitly) made aware of the underlying structure does not trivialize this effect. Rather, our results indicate that the continuous stream of input was correctly parsed and the underlying linguistic structure encoded despite being task-irrelevant.

6. Speech manipulation:• To assist interpretation of the pattern of results it would be helpful to know something about the listening experience to the stimuli. Was there any check on abnormality of the attended speech that was changed in gender, i.e., how natural did this sound? Why was the gender manipulation imposed, instead of initially recording the stimuli in a male voice?

The natural speech materials were chosen from a pre-existing database of actor-recorded short stories, that have been used in previous studies in the lab. These were originally recorded by both female and male speakers. However, our frequency-tagged stimuli were recorded only in a male voice. Since it is known that selective attention to speech is highly influenced by whether the voices are of the same/different sex, we opted to use only male voices. Therefore, we used the voice-change transformation on narratives that were originally spoken by a female actor. To ensure that the gender change did not affect the natural sounding of the speech and to check for abnormalities of the materials, we conducted a short survey among 10 native Hebrew speakers. They all agreed that the speech sounded natural and normal. Examples of the stimuli are now available at: https://osf.io/e93qa. We have clarified this in the methods section (pp. 6) which reads:

“Natural speech stimuli were narratives from publicly available Hebrew podcasts and short audio stories (duration: 44.53±3.23 seconds). […] They all agreed that the speech sounded natural and normal.”• Similarly, how natural-sounding was the task-irrelevant structured speech?

The task-irrelevant speech is comprised of individually-recorded syllables, concatenated at a fixed rate. Although the syllabic-rate (4Hz) is akin to that of natural speech, the extreme rhythmicity imposed here is not natural and requires some perceptual adaptation. As mentioned above, the frequency-tagged stimuli are not immediately recognizable as Hebrew speech, which is the main reason for conducting a familiarization stage prior. After a few minutes of familiarization these stimuli still sound unnaturally-rhythmic, but are fully intelligible as Hebrew. Examples of the stimuli (presented separately and dichotically) are now available at: https://osf.io/e93qa.

7. Behavioral Results:• It is difficult to reconcile why there are no significant differences in response accuracy between non-structured and structured conditions. How do the authors interpret this discrepancy alongside the observed neural differences across these two conditions? Is it possible this is reflective of the stimulus, i.e., that the structured speech is so artificial that it is not intrusive on the attended signal (thus no difference in behavior)? It seems less plausible that the non-structured speech was equally as intrusive as the structured speech, given the cited literature. These issues relate to the speech manipulation.• Did the authors check on task behavior before running the experiment?• It would be helpful to understand why no behavioral result occurred, and whether this was related to the speech manipulation specifically by obtaining behavioral results with other versions of the task.

See our extensive answer above regarding the behavioral tasks (including results from behavioral screening of the task) in response to critique #2.

• How does this relate to the irrelevant stimuli were used in the experiment in Makov et al. 2017. Is it possible this is important, can the authors comment?

Yes. The Structured stimuli were identical to those used in our previous study by Makov et al., 2017. However, in that study only a single-speech stimulus was presented and selective attention was not manipulated. We now mention this explicitly in the methods section.

• There is a rather large range in response accuracy across subjects. One analysis that might strengthen the argument that the increase in 1 Hz ITPC truly reflects phrasal level information would be to look for a correlation between response accuracy in the structured condition with 1 Hz ITPC. One might predict that listeners which show lower behavioral accuracy would show greater 1 Hz ITPC (i.e. greater interference from linguistic content of the unattended structured speech).

Thank you for this suggestion. To test whether the 1Hz ITPC peak was stronger in participants who performed poorly we have added a new analysis. We split the participants into two groups according to the median of the 1Hz ITPC value in the Structured condition and tested whether there was a significant difference in the behavioral scores of the two groups. However, this median-split analysis did not reveal any significant effects that would indicate a ‘trade-off’ between linguistic representation of task-irrelevant speech and performance on the attended task. That said, as elaborated in our response to critique #2 above, the task was not sufficiently sensitive for adequately capturing behavioral consequences of task-irrelevant stimuli. Therefore, in our opinion, this null-effect should not be taken as indication that both types of task-irrelevant stimuli were similarly distracting.

We have added this new analysis to the paper, which reads (p. 13-14):

“Additionally, to test for possible interactions between answering questions about the to-be attended speech and linguistic neural representation of task-irrelevant speech, we performed a median-split analysis of the behavioral scores across participants. […] Neither test showed significant differences in performance between participants whose 1Hz ITPC was above vs. below the median [Structured condition: t(27) = -1.07, p=0.29; Structured – Non-Structured: t(27) = -1.04, p=0.15]. Similar null-results were obtained when the median-split was based on the source level data.”8. Attention:• An attentional load account is interesting, but load is not directly manipulated in this experiment. And it is difficult to reconcile the claim that the Ding experiment was less natural than the current one, as both were transformations of time.

Thank you for this opportunity to clarify our comparison between the current study and the study by Ding et al.: While both studies used similar frequency-tagged speech as task irrelevant stimuli, the to-be-attended stimuli used by us was natural speech whereas Ding et al. used speech that was compressed by a factor of 2.5 and where all naturally-occurring gaps were artificially removed. **I**t is well known that understanding time-compressed speech is substantially more difficult than natural-paced speech. Hence, in order to perform the task (answer questions about the to-be-attended narrative), participants in the study by Ding et al. needed to invest substantially more listening effort relative to the current one.

We offer the perspective of “Attentional Load theory” as a way to account for the discrepancy in results between these two studies, and also as a way to think more broadly about the discrepancies reported throughout the literature regarding processing of task-irrelevant stimuli. Specifically, we proposed that because of the more-demanding stimuli/task used by Ding et al., insufficient resources were available to also encode task-irrelevant stimuli, which is in line with “Attentional Load Theory” (Lavie et al. 2004). We also agree that additional studies are required in order to fully test this explanation and systematically manipulate “load” in a within-experiment design. We have revised our discussion on this point (pp. 21):

“While in the current experiment to-be-attended speech was presented in its natural form, mimicking the listening effort of real-life speech-processing, in the study by Ding et al. (Ding et al. 2018) to-be-attended speech was time-compressed by a factor of 2.5 and naturally occurring gaps were removed, making the comprehension task substantially more effortful (Nourski et al. 2009; Müller et al. 2019). […] Specifically, if understanding the to-be-attended speech imposes relatively low perceptual and cognitive load, then sufficient resources may be available to additionally process aspects of task-irrelevant speech, but that this might not be the case as the task becomes more difficult and perceptually demanding (Wild et al. 2012; Gagné et al. 2017; Peelle 2018).”• Federenko and Blank, 2020 propose an account of inferior frontal mechanisms that may relate to the present pattern of results regarding, in particular that there may be a stronger manipulation of attention (domain general) than linguistic processes.

We thank the reviewer for pointing us to this highly-relevant work, and now discuss the potential involvement of IFG in domain-general processes such as working-memory and attention, besides its role in speech processing (pp. 23).

9. Lack of word level response:• Again it might be relevant with respect to potential stimulus differences to previous versions of the stimuli. If this is the case then it might have important implications for only a 1Hz effect being found.• If SNR is an issue it calls into question the current results in addition to the possibility that it was an underestimation.

See our extensive answer regarding the lack of a work-level peak in response to critique #4 above, where we address the lack of a word-level response and include a new analysis validating the robustness of the 1Hz ITPC peak.

10. Familiarization phase:• If they exist, what are the behavioral data / accuracy between conditions (structured v. non-structured) from the familiarization phase? Or did subjects comment on any differences?

See our answer above regarding the familiarization task, in response to critique #5.

11. ROIs:• A representation or list of the ROIs might be helpful. It is unclear from Figure 1 if the whole cortical mantle is included in the 22 ROIs. In addition these ROI boundaries look considerably larger than the 180 in each hemisphere from Glasser et al. 2016. Please clarify.

We apologize, this was not clear enough in our original submission. Indeed, Glasser et al., 2016, identified 180 ROIs in each hemisphere. However, in the Supplementary Neuroanatomical Results (table1 p180) these are grouped into 22 larger ROIs, which is extremely useful for data-simplification and reduction of multiple-comparisons. This coarser ROI-division is also more suitable for MEG data, given its reduced spatial-resolution (compared to fMRI). The 22 ROIs are delineated in black in Figure 3, and we have now clarified this in the methods section (pp. 12)

Reviewer #2:[…] The only issue I had with the results is that the possibility (or likelihood, in my estimation) that the subjects are occasionally letting their attention drift to the task-irrelevant speech rather than processing in parallel can't be rejected. To be fair, the authors include a nice discussion of this very issue and are careful with the language around task-relevance and attended/unattended stimuli. It is indeed tough to pull apart. The second paragraph on page 18 states "if attention shifts occur irregularly, the emergence of a phase-rate peak in the neural response would indicate that bits of 'glimpsed' information are integrated over a prolonged period of time." I agree with the math behind this, but I think it would only take occasional lapses lasting 2 or 3 seconds to get the observed results, and I don't consider that "prolonged." It is, however, much longer than a word, so nicely rejects the idea of single-word intrusions.

We fully agree with the reviewer that the current results do not necessarily imply parallel processing of competing speech and could be brought about by occasional (rhythmic or nonrhythmic) shifts of attention between streams (“multiplexed listening”). We now elaborate on our previous discussion of this issue, emphasizing two main points (p. 19-20; also, please see our extensive response to reviewer #1/critique #1):

1. As the reviewer points out, regardless of the underlying mechanism, the current results indicate temporal integration of content and structure-building processing, that go beyond momentary “intrusions” of single words.

2. One of our main take-home messages from the current study is that selective-attention does not mean exclusive-attention, and that individuals are capable of gleaning linguistic components of task-irrelevant speech (be it through parallel processing or through multiplexed-listening). This invites us to re-examine some of the assumptions regarding what individuals are actually *doing* when we *instruct* them to pay selective attention.

Reviewer #3:The use of frequency tagging to analyze continuous processing at phonemic, word, phrasal and sentence-levels offers a unique insight into neural locking at higher-levels. While the approach is novel, there are major concerns regarding the technical details and interpretation of results to support phrase-level responses to structured speech distractors.– Is the peak at 1Hz real and can it be attributed solely to the structured distractor?* The study did not comment on the spectral profile of the "attended" speech, and how much low modulation energy is actually attributed to the prosodic structure of attended sentences? To what extent does the interplay of the attended utterance and distractor shapes the modulation dynamics of the stimulus (even dichotically)?

The reviewer suggests that, perhaps, the acoustics of the task-relevant stream could have contributed to the observed 1Hz ITPC peak, and that the peak might not solely be attributed to the Structured stimuli.

There are two reasons why this is not the case.

1. In Author response image 1 we show the modulation spectrum of the narratives used as task-relevant speech. As can be seen, the spectrum does not contain a distinct peak at 1Hz.

2. The narratives used as task-relevant speech were fully randomized between the Structured and Non-Structured conditions and between participants. Therefore, if these stimuli had contributed to the 1Hz peak, this should have been observed in both conditions. We have now clarified this point in the methods (p.6) which reads “For each participant they were randomly paired with task-irrelevant speech (regardless of condition), to avoid material-specific effects.”

**Author response image 1. sa2fig1:** 

* How is the ITPC normalized? Figure 2 speaks of a normalization but it is not clear how? The peak at 1Hz appears extremely weak and no more significant (visually) than other peaks – say around 3Hz and also 2.5Hz in the case of non-structured speech? Can the authors report on the regions in modulation space that showed any significant deviations? What about effect size of the 1Hz peak relative to these other regions?* It is hard to understand where the noise floor in this analysis – this floor will rotate with the permutation test analysis performed in the analysis of the ITPC and may not be fully accounted for. This issue depends on what the chosen normalization procedure is. The same interpretation put forth by the author regarding a lack of a 0.5Hz peak due to noise still raises the question of interpreting the observed 1Hz peak?

We understand the reviewer’s concerns about the validity of the 1Hz peak, and have taken several steps to address them:

1. We have added **a new analysis to independently assess the significance of the 1Hz peak. In this a**nalysis, we compared ITPC at each frequency to the average ITPC in the surrounding frequencies (2 bins from each side) using a t-test. This approach addresses the concern of frequency-specific noise-floor, for example due to the inherent 1/f noise structure (see similar implementations e.g., in Mouraux et al., 2011; Retter and Rossion, 2016, Nozaradan et al. 2017, 2018). Results of this analysis confirmed that the only significant ITPC peaks were at 4Hz (in both conditions) and at 1Hz (in the Structured condition only), further validating the robustness of the 1Hz phrase-level response. Note that the other peaks that stand-out visually (at 2.5Hz and 3Hz) were not statistically significant in this analysis. The new analysis is now described in the methods (pp. 11) and Results sections (pp. 14) which reads:

“Scalp-level spectra of the Inter-trial phase coherence (ITPC) showed a significant peak at the syllabic-rate (4Hz) in response to both Structured and Non-Structured hierarchical frequency-tagged speech, with a 4-pole scalp-distribution common to MEG recorded auditory responses (Figure 3a) (p< 10^-9; large effect size, Cohen's d > 1.5 in both). […] Comparison of the 1Hz ITPC between these conditions also confirmed a significant difference between them (p=0.045; moderate effect size, Cohen's d = 0.57).”

2. Although, as the reviewer points out, the magnitude of the 1Hz peak is smaller than the 4Hz, this is to be expected since the 4Hz peak is a direct consequence of the acoustic input whereas the 1Hz peak proposedly reflects linguistic representation and/or chunking. Similar ratios between the acoustic and linguistic peaks have been observed in previous studies using this frequency-tagging approach (e.g. Ding et al. 2016, Makov et al. 2017). Therefore, the smaller relative magnitude of the 1Hz peak does not invalidate it. We have now added quantification of all effect sizes to the Results section.

3. The normalization used for the ITPC was a z-score of the resultant phase-locking factor, as implemented in the circ_rtest function in the Matlab circular statistics toolbox (Berens, 2009). We have now clarified this in the methods section (p. 11) and the caption of Figure 3 (p. 14).

2. Control of attention during task performance* The author present a very elegant analysis of possible alterative accounts of the results, but they acknowledge that possible attention switches, even if irregular, could result in accumulated information that could emerge as a small neurally-locked response at the phrase-level? As indicated by the authors, the entire experimental design to fully control for such switches is a real feat. That being said, additional analyses could shed some light on variations of attentional state and their effect on observed results. For instance, analysis of behavioral data across different trials (wouldn't be conclusive, but could be informative).

Indeed, we agree that occasional ‘attention switches’ can be one explanation for the results observed here. We now elaborate on our previous discussion of this point in the revised paper (discussion, pp. 20; and see also our extensive response to reviewer #1/critique #1 on this point).

To address the reviewer’s suggestion of looking more closely at the behavioral results, and testing whether the emergence of a phrase-level response to task-irrelevant speech came at the ‘expense’ of processing the to-be-attended speech, we now include a new median-split analysis. Rather than conducting a per-trial analysis (which suffers from low SNR), we split the participants into two groups according to the median of the 1Hz ITPC value in the Structured condition and tested whether there was a significant difference in the behavioral scores of the two groups. However, this median-split analysis did not reveal any significant effects that would indicate a ‘tradeoff’ between linguistic representation of task-irrelevant speech and performance on the attended task. We have added this new analysis to the paper, which reads (p. 13-14):

“Additionally, to test for possible interactions between answering questions about the to-be attended speech and linguistic neural representation of task-irrelevant speech, we performed a median-split analysis of the behavioral scores across participants. […] Neither test showed significant differences in performance between participants whose 1Hz ITPC was above vs. below the median [Structured condition: t(27) = -1.07, p=0.29; Structured – Non-Structured: t(27) = -1.04, p=0.15]. Similar null-results were obtained when the median-split was based on the source level data.”

That said, as elaborated in our response to reviewer #1/critique #2 above, and in the revised methods (p. 9) we do not believe the task used here is sufficiently sensitive for capturing behavioral consequences of task-irrelevant stimuli. The lack of a good behavioral measures is, in fact, part of the challenge of studying attention to continuous speech and part of the motivation for turning towards neural metrics rather than relying on behavioral measures, as we now discuss explicitly (p. 25):

“In moving towards studying speech processing and attention under more ecological circumstances, using natural continuous speech, we face an experimental challenge of obtaining sufficiently sensitive behavior measures without disrupting listening with an ongoing task (e.g., target detection) or encroaching too much on working-memory. This is a challenge shared by many previous studies similar to ours, and is one of the main motivations for turning directly to the brain and studying neural activity during uninterrupted listening to continuous speech, rather than relying on sparse behavioral indications (Ding et al. 2016; Makov et al. 2017; Broderick et al. 2018, 2019; Donhauser and Baillet 2019; Brodbeck et al. 2020).”

Therefore, although we wholeheartedly agree with the reviewer’s comment that “the experimental design to fully control for such switches is a real feat”, and we DO hope to be able to better address this question in future (neural) studies, in our opinion the current behavioral data cannot reliably shed light on this matter.

* This issue is further compounded by the fact that a rather similar study (Ding et al.) did not report any phrasal-level processing, though there are design differences. The authors suggest differences in attentional load as a possible explanation and provide a very appealing account or reinterpretation of the literature based on a continuous model of processing based on task demands. While theoretically interesting, it is not clear whether any of the current data supports such account. Again, maybe a correlation between neural responses and behavioral performance in specific trials could shed some light or strengthen this claim.

Here too, we refer the reviewer to our response to reviewer #1/critique #8, regarding the comparison between the current results and the study by Ding et al.

We offer the perspective of “Attentional Load theory” as a way to account for the discrepancy in results between these two studies, given that they differ substantially in the listening effort required for performing the task. We agree with the reviewer that, at the moment, this is mostly a theoretical speculation (particularly given the null-behavioral effects discussed above), even though it does converge with several other lines of empirical testing (e.g. the effects of working-memory capacity on attentional abilities). It also gives rise to specific hypotheses which can (and should) be tested in follow-up studies, which is why we think it is beneficial to include this as a discussion point in our paper (p. 21-22):

“the comparison between these two studies invites re-framing of the question regarding the type / level of linguistic processing applied to task-irrelevant speech, and propels us to think about this issue not as a yes-or-no dichotomy, but perhaps as a more flexible process that depends on the specific context (Brodbeck et al. 2020b). […] As cognitive neuroscience research increasingly moves towards studying speech processing and attention in real-life circumstances, a critical challenge will be to systematically map out the perceptual and cognitive factors that contribute to, or hinder, the ability to glean meaningful information from stimuli that are outside the primary focus of attention.”– What is the statistic shown for the behavioral results? Is this for the multiple choice question? Then what is the t-test on?

We are happy to clarify. After each narrative, participants were asked 4 multiple-choice questions (with three potential answers; chance level = 0.33). Their answers to question were coded as ‘correct’ or ‘incorrect’ and then the average accuracy-rate (% questions answered correctly) was calculated across all questions and all narratives, for each participant. This was done separately for trials in the Structured and Non-Structured condition.

The t-test presented in Figure 2 tests for differences in average-accuracy rate between conditions. Additionally, we performed a t-test between the accuracy rate averaged across all conditions vs. chance-rate (0.33), to establish that they performed better than chance (guessing).

We have now clarified this in the methods section (p. 9):

“The average accuracy rate of each participant (% questions answered correctly) was calculated across all questions and narratives, separately for trials in the Structured and NonStructured condition.”– Beyond inter-trial phase coherence, can the authors comment on actual power-locked responses at the same corresponding rates?

Thank you for this comment. Yes, we did look at the evoked power-spectrum in addition to the ITPC (Author response image 2). As is evident, the power spectrum also contained a clearly visible peak at 1Hz in the Structured condition (but *not* in the Non-Structured condition). However, statistical testing of this peak vs. its surrounding neighbors was not significant. We attribute the discrepancy between the power spectrum and ITPC spectrum to the 1/f power-law noise than primarily affects the former (attributed to non-specific/spontaneous neural activity; e.g. Voytek et al. 2015).

Similar patterns have also been observed in other studies using frequency-tagged stimuli, and ITPC seems to be a ‘cleaner’ measure for bringing out frequency-specific phase-locked responses in the neural signal (see similar discussion in Makov et al. 2017). For this reason, in the manuscript we only report the ITPC results.

In line with concerns regarding interpretation of experimental findings, some control experiment appears to be critical to establish a causal link between the observed neural processing the 1Hz rhythm and the phrasal processing of the distractor. What do the author expect from a similarly structured distractor but unintelligible, in an unfamiliar language or even reversed?

Multiple previous studies using hierarchical frequency-tagging of speech have clearly shown that this response is specific to contexts in which individuals understand the speech. For example, both Ding et al. 2016 and Makov et al. 2017 showed that the phrase-level response is only produced for speech in a familiar language, but not for unfamiliar language or random concatenation of syllables. This pattern of results has been replicated across several languages so far (e.g., English, Chinese, Hebrew, Dutch) as well as for newly learned pseudo-languages (Henin et al. 2021). Accordingly, the current study builds on these previous findings in attributing the 1Hz peak to encoding the phrase-structure of speech.